# INVERTIBLE GENERATIVE MODELS FOR INVERSE PROBLEMS: MITIGATING REPRESENTATION ERROR AND DATASET BIAS

## ABSTRACT

Trained generative models have shown remarkable performance as priors for inverse problems in imaging. For example, Generative Adversarial Network priors permit recovery of test images from 5-10x fewer measurements than sparsity priors. Unfortunately, these models may be unable to represent any particular image because of architectural choices, mode collapse, and bias in the training dataset. In this paper, we demonstrate that invertible neural networks, which have zero representation error by design, can be effective natural signal priors at inverse problems such as denoising, compressive sensing, and inpainting. Our formulation is an empirical risk minimization that does not directly optimize the likelihood of images, as one would expect. Instead we optimize the likelihood of the latent representation of images as a proxy, as this is empirically easier. For compressive sensing, our formulation can yield higher accuracy than sparsity priors across almost all undersampling ratios. For the same accuracy on test images, they can use 10-20x fewer measurements. We demonstrate that invertible priors can yield better reconstructions than sparsity priors for images that have rare features of variation within the biased training set, including out-of-distribution natural images.

## 1 INTRODUCTION

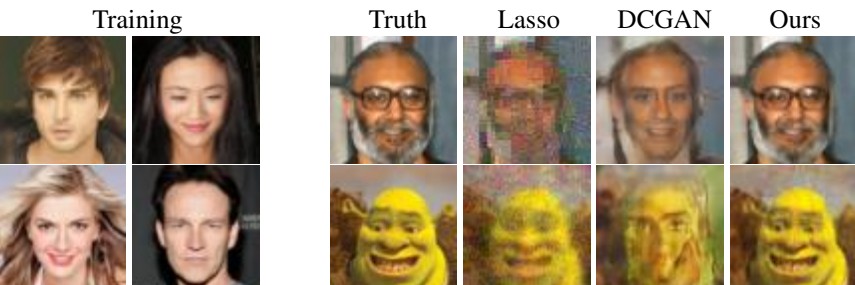

Figure 1: We train an invertible generative model with CelebA images (including those at left). When used as a prior for compressed sensing, it can yield higher quality image reconstructions than Lasso and a trained DCGAN, even on out-of-distribution images. Note that the DCGAN reflects biases of the training set by removing the man's glasses and beard, whereas our invertible prior does not.

Generative deep neural networks have shown remarkable performance as natural signal priors in imaging inverse problems, such as denoising, inpainting, compressed sensing, blind deconvolution, and phase retrieval. These generative models can be trained from datasets consisting of images of particular natural signal classes, such as faces, fingerprints, MRIs, and more (Karras et al., 2017; Minaee and Abdolrashidi, 2018; Shin et al., 2018; Chen et al., 2018). Some such models, including variational autoencoders (VAEs) and generative adversarial networks (GANs), learn an explicit low-dimensional manifold that approximates a natural signal class (Goodfellow et al., 2014; Kingma and Welling, 2013; Rezende et al., 2014). We will refer to such models as GAN priors. With an explicit parameterization of the natural signal manifold by a low dimensional latent representation,

these generative models allow for direct optimization over a natural signal class. Consequently, they can obtain significant performance improvements over non-learning based methods. For example, GAN priors have been shown to outperform sparsity priors at compressed sensing with 5-10x fewer measurements. Additionally, GAN priors have led to theory for signal recovery in the linear compressive sensing and nonlinear phase retrieval problems (Bora et al., 2017; Hand and Voroninski, 2017; Hand et al., 2018), and they have also shown promising results for the nonlinear blind image deblurring problem (Asim et al., 2018).

A significant drawback of GAN priors for solving inverse problems is that they can have representation error or bias due to architecture and training. This can happen for many reasons, including because the generator only approximates the natural signal manifold, because the natural signal manifold is of higher dimensionality than modeled, because of mode collapse, or because of bias in the training dataset itself. As many aspects of generator architecture and training lack clear principles, representation error of GANs may continue to be a challenge even after substantial hand crafting and engineering. Additionally, learning-based methods are particularly vulnerable to the biases of their training data, and training data, no matter how carefully collected, will always contain degrees of bias. As an example, the CelebA dataset (Liu et al., 2015) is biased toward people who are young, who do not have facial hair or glasses, and who have a light skin tone. As we will see, a GAN prior trained on this dataset learns these biases and exhibits image recovery failures because of them.

In contrast, invertible neural networks can be trained as generators with zero representation error. These networks are invertible (one-to-one and onto) by architectural design (Dinh et al., 2016; Gomez et al., 2017; Jacobsen et al., 2018; Kingma and Dhariwal, 2018). Consequently, they are capable of recovering any image, including those significantly out-of-distribution relative to a biased training set; see Figure 1. We call the domain of an invertible generator the latent space, and we call the range of the generator the signal space. These must have equal dimensionality. Flow-based invertible generative models are composed of a sequence of learned invertible transformations. Their strengths include: their architecture allows exact and efficient latent-variable inference, direct log-likelihood evaluation, and efficient image synthesis; they have the potential for significant memory savings in gradient computations; and they can be trained by directly optimizing the likelihood of training images. This paper emphasizes an additional strength: *because they lack representation error, invertible models can mitigate dataset bias and improve performance on inverse problems with out-of-distribution data.*

In this paper, we study generative invertible neural network priors for imaging inverse problems. We will specifically use the Glow architecture, though our framework could be used with other architectures. A Glow-based model is composed of a sequence of invertible affine coupling layers, 1x1 convolutional layers, and normalization layers. Glow models have been successfully trained to generate high resolution photorealistic images of human faces (Kingma and Dhariwal, 2018).

We present a method for using pretrained generative invertible neural networks as priors for imaging inverse problems. The invertible generator, once trained, can be used for a wide variety of inverse problems, with no specific knowledge of those problems used during the training process. Our method is an empirical risk formulation based on the following proxy: we penalize the likelihood of an image's latent representation instead of the image's likelihood itself. While this may be couterintuitive, it admits optimization problems that are easier to solve empirically. In the case of compressive sensing, our formulation succeeds even without direct penalization of this proxy likelihood, with regularization occuring through initialization of a gradient descent in latent space.

We train a generative invertible model using the CelebA dataset. With this fixed model as a signal prior, we study its performance at denoising, compressive sensing, and inpainting. For denoising, it can outperform BM3D (Dabov et al., 2007). For compressive sensing on test images, it can obtain higher quality reconstructions than Lasso across almost all subsampling ratios, and at similar reconstruction errors can succeed with 10-20x fewer measurements than Lasso. It provides an improvement of about 2x fewer linear measurements when compared to Bora et al. (2017). Despite being trained on the CelebA dataset, our generative invertible prior can give higher quality reconstructions than Lasso on out-of-distribution images of faces, and, to a lesser extent, unrelated natural images. Our invertible prior outperforms a pretrained DCGAN (Radford et al., 2015) at face inpainting and exhibits qualitatively reasonable results on out-of-distribution human faces. We provide additional experiments in the appendix, including for training on other datasets.

## 2 METHOD AND MOTIVATION

We assume that we have access to a pretrained generative invertible neural network $G : \mathbb{R}^n \to \mathbb{R}^n$. We write $x = G(z)$ and $z = G^{-1}(x)$, where $x \in \mathbb{R}^n$ is an image that corresponds to the latent representation $z \in \mathbb{R}^n$. We will consider a $G$ that has the Glow architecture introduced in Kingma and Dhariwal (2018). It can be trained by direct optimization of the likelihood of a collection of training images of a natural signal class, under a standard Gaussian distribution over the latent space. We consider recovering an image $x$ from possibly-noisy linear measurements given by $A \in \mathbb{R}^{m \times n}$,

$$y = Ax + \eta,$$

where $\eta \in \mathbb{R}^m$ models noise. Given a pretrained invertible generator $G$, we have access to likelihood estimates for all images $x \in \mathbb{R}^n$. Hence, it is natural to attempt to solve the above inverse problem by a maximum likelihood formulation given by

$$\min_{x \in \mathbb{R}^n} \|Ax - y\|^2 - \gamma \log p_G(x), \tag{1}$$

where $p_G$ is the likelihood function over $x$ induced by $G$, and $\gamma$ is a hyperparameter. We have found this formulation to be empirically challenging to optimize; hence we study the following proxy:

$$\min_{z \in \mathbb{R}^n} \|AG(z) - y\|^2 + \gamma \|z\|. \tag{2}$$

Unless otherwise stated, we initialize (2) at $z_0 = 0$.

The motivation for formulation (2) is as follows. As a proxy for the likelihood of an image $x \in \mathbb{R}^n$, we will use the likelihood of its latent representation $z = G^{-1}(x)$. Because the invertible network $G$ was trained to map a standard normal in $\mathbb{R}^n$ to a distribution over images, the log-likelihood of a point $z$ is proportional to $\|z\|^2$. Instead of penalizing $\|z\|^2$, we alternatively penalize the unsquared $\|z\|$. In Appendix B, we show comparable performance for both the squared and unsquared formulations.

In principle, our formulation has an inherent flaw: some high-likelihood latent representations $z$ correspond to low-likelihood images $x$. Mathematically, this comes from the Jacobian term that relates the likelihood in $z$ to the likelihood in $x$ upon application of the map $G$. For multimodel distributions, such images must exist, which we will illustrate in the discussion. This proxy formulation relies on the fact that the set of such images has low probability and that they are inconsistent with enough provided measurements. Surprisingly, despite this potential weakness, we will observe image reconstructions that are superior to BM3D and GAN-based methods at denoising, and superior to GAN-based and Lasso-based methods at compressive sensing.

In the case of compressive sensing and inpainting, we take $\gamma = 0$ in formulation (2). The motivation for such a formulation initialized at $z_0 = 0$ is as follows. There is a manifold of images that are consistent with the provided measurements. We want to find the image $x$ of highest likelihood on this manifold. Our proxy turns the likelihood maximization task over an affine space in $x$ into the geometric task of finding the point on a manifold in $z$-space that is closest to the origin with respect to the Euclidean norm. In order to approximate that point, we run a gradient descent in $z$ down the data misfit term starting at $z_0 = 0$.

In the case of GAN priors for $G : \mathbb{R}^k \to \mathbb{R}^n$, we will use the formulation from Bora et al. (2017), which is the formulation above in the case where the optimization is performed over $\mathbb{R}^k$, $\gamma = 0$, and initialization is selected randomly.

All the experiments that follow will be for an invertible model we trained on the CelebA dataset of celebrity faces, as in Kingma and Dhariwal (2018). Similar results for models trained on birds and flowers (Wah et al., 2011; Nilsback and Zisserman, 2008) can be found in the appendix. Due to computational considerations, we run experiments on $64 \times 64$ color images with the pixel values scaled between $[0, 1]$. The train and test sets contain a total of 27,000 and 3,000 images, respectively. We trained a Glow architecture (Kingma and Dhariwal, 2018); see Appendix A for details. Once trained, the Glow prior is fixed for use in each of the inverse problems below. We also trained a DCGAN for the same dataset. We solve (2) using LBFGS, which was found to outperform Adam (Kingma and Ba, 2014). DCGAN results are reported for an average of 3 runs because we observed some variance due to random initialization.

# 3 APPLICATIONS

## 3.1 DENOISING

We consider the denoising problem with $A = I$ and $\eta \sim \mathcal{N}(0, \sigma^2 I)$, for images $x$ in the CelebA test dataset. We evaluate the performance of a Glow prior, a DCGAN prior, and BM3D for two different noise levels. Figure 2 shows the recovered PSNR values as a function of $\gamma$ for denoising by the Glow and DCGAN priors, along with the PSNR by BM3D. The figure shows that the performance of the regularized Glow prior increases with $\gamma$, and then decreases. If $\gamma$ is too low, then the network fits to the noise in the image. If $\gamma$ is too high, then data fit is not enforced strongly enough. The left panel reveals that an appropriately regularized Glow prior can outperform BM3D by almost 2 dB. The experiments also reveal that appropriately regularized Glow priors outperform the DCGAN prior, which suffers from representation error and is not aided by the regularization. The right panel confirms that with smaller noise levels, less regularization is needed for optimal performance. A

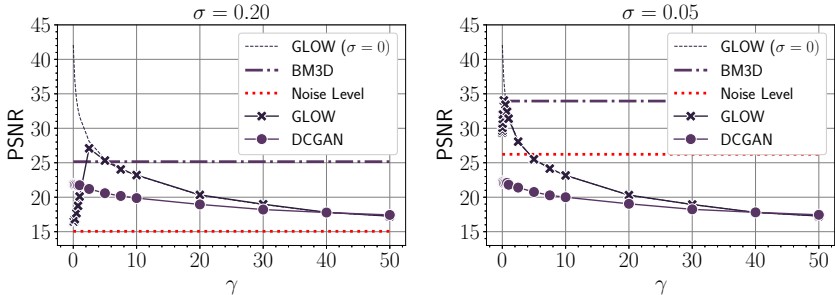

Figure 2: Recovered PSNR values as a function of $\gamma$ for denoising by the Glow and DCGAN priors. All the results are averaged over 12 test set images. For reference, we show the average PSNRs of the original noisy images, after applyig BM3D, and under the Glow prior in the noiseless case ($\sigma = 0$).

visual comparison of the recoveries at the noise level $\sigma = 0.1$ using Glow, DCGAN priors, and BM3D can be seen in Figure 3. Note that the recoveries with Glow are sharper than BM3D. See Appendix B for more quantitative and qualitative results.

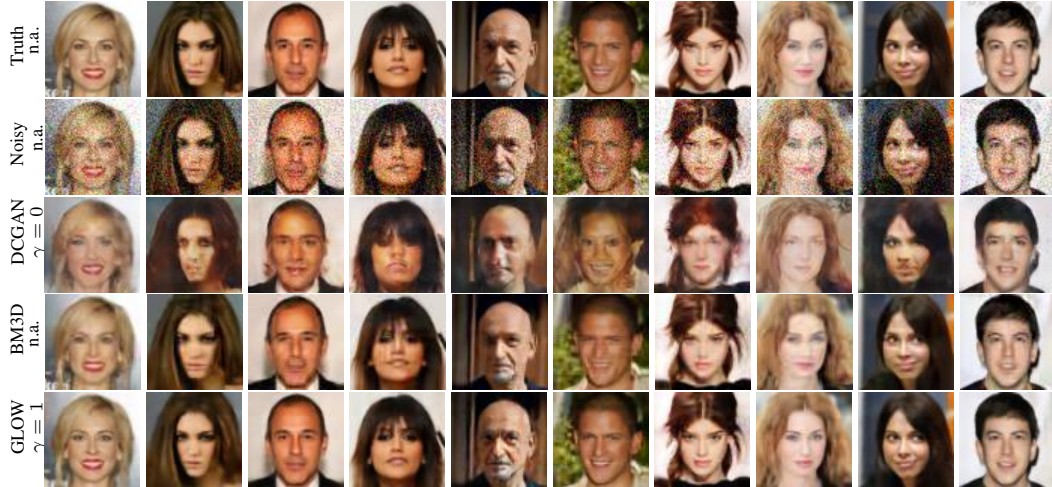

Figure 3: Denoising results using the Glow prior, the DCGAN prior, and BM3D at noise level $\sigma = 0.1$. Note that the Glow prior gives a sharper image than BM3D.

## 3.2 COMPRESSED SENSING

In compressed sensing, one is given undersampled linear measurements of an image, and the goal is to recover the image from those measurements. In our notation, $A \in \mathbb{R}^{m \times n}$ with $m < n$. As the

image $x$ is undersampled, there is an affine space of images consistent with the measurements, and an algorithm must select which is most 'natural.' A common proxy for naturalness in the literature has been sparsity with respect to the DCT or wavelet bases. With a GAN prior, an image is considered natural if it lies in or near the range of the GAN. For an invertible prior under our proxy for likelihood, we consider an image to be natural if it has a latent representation of small norm.

We study compressed sensing in the case that $A$ is an $m \times n$ matrix of i.i.d. $\mathcal{N}(0, 1/m)$ entries, and $x$ is an image from the CelebA test set. Here, $n = 64 \times 64 \times 3 = 12288$. We consider the case where $\eta$ is standard iid Gaussian random noise normalized such that $\sqrt{\mathbb{E}\|\eta\|^2} = 0.1$. We compare Glow, DCGAN, and Lasso[1] with respect to the DCT and wavelet bases.

Our main result is that the Glow prior with $\gamma = 0$ and initialization $z_0 = 0$ outperforms both DCGAN and Lasso in reconstruction quality over all undersampling ratios, as shown in the left panel of Figure 4. Surprisingly, in the case of extreme undersampling, Glow substantially outperforms these methods even though it does not maintain a direct low-dimensional parameterization of the signal manifold. The Glow prior (1) can result in 15 dB higher PSNRs than DCGAN, and (2) can give comparable recovery errors with 2-3x fewer measurements at high undersampling ratios. This difference is explained by the representation error of DCGAN, which has been shown to be the dominant source of error in DCGAN by Bora et al. (2017). Additional plots and visual comparisons, available in Appendix C, show notable improvements in quality of in- and out-of-distribution images using an invertible prior relative to DCGAN and Lasso.

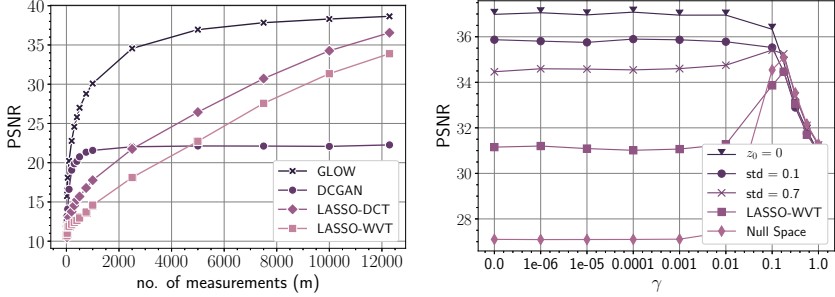

Figure 4: The left panel shows recovered PSNRs averaged over 12 test set images under the Glow, and DCGAN prior with $\gamma = 0$; and the Lasso with respect to the DCT and a Wavelet Transform. We initialize with $z_0 = 0$. See Appendix C for a zoom-in of the case of small $m$. The right panel shows the resulting PSNR when $m = 5000$ with a Glow prior after different initialization strategies, as described in the text. The highest PSNR was recovered with initialization $z_0 = 0$ and $\gamma = 0$.

We conducted several additional experiments to understand the regularizing effects of $\gamma$ and the initialization $z_0$. The right panel of Figure 4 shows the PSNRs under multiple initialization strategies: $z_0 = 0$, $z_0 \sim \mathcal{N}(0, 0.1^2 I)$, $z_0 \sim \mathcal{N}(0, 0.7^2 I)$, $z_0 = G^{-1}(x_0)$ with $x_0$ given by the solution to Lasso with respect to the wavelet basis, and $z_0 = G^{-1}(x_0)$ where $x_0$ is $x$ perturbed by a random point in the null space of $A$. The best performance was observed with initialization $z_0 = 0$. The hyperparameter $\gamma$ can be taken to be zero, which is surprising because then there is no direct penalization of likelihood for this noisy compressive sensing problem. In the case of $\gamma = 0$, we observe that larger initializations result in recovered images of lower PSNR. See Appendix C for additional experiments that show this effect. We observe that initialization strategy can have a strong qualitative effect on the recovery formulation. For example, if the optimization is initialized by the solution to the Lasso, then directly penalizing the likelihood of $z$ can improve reconstruction PSNR, though those reconstruction are still worse than with initialization $z_0 = 0$ and $\gamma = 0$. Suboptimal initialization procedures apparently benefit from direct penalization of likelihood, whereas the $z_0 = 0$ initialization apparently does not.

Finally, we observe that the Glow prior is much more robust to out-of-distribution examples than the GAN Prior. Figure 5 shows recovered images using (2) for compressive sensing for images not belonging to the CelebA dataset. DCGAN's performance reveals biases of the underlying dataset and limitations of low-dimensional modeling. For example, projecting onto the CelebA-trained DCGAN can cause incorrect skin tone, gender, and age. It's performance on out-of-distribution images is poor.

---

[1]The inverse problems with Lasso were solved by $\min_z \|A\Phi z - y\|_2^2 + 0.01\|z\|_1$ using coordinate descent.

In contrast, the Glow prior mitigates this bias, even demonstrating image recovery for natural images that are not representative of the CelebA training set, including people who are older, have darker skin tones, wear glasses, have a beard, or have unusual makeup. The Glow prior's performance also extends to significantly out-of-distribution images, such as animated characters and natural images unrelated to faces. See Appendix C.2 for additional experiments.

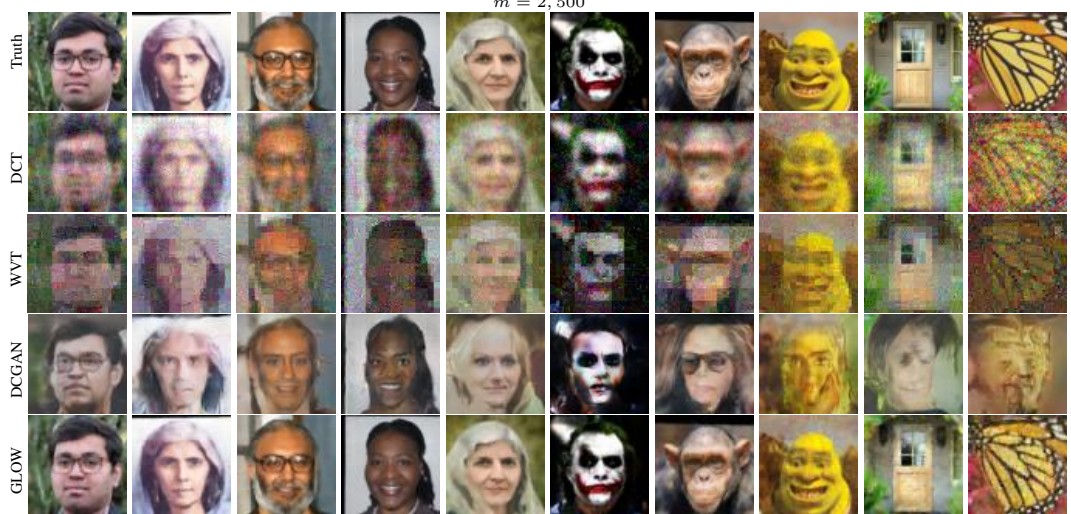

Figure 5: Compressed sensing (CS) with a number $m = 2,500$ ($\approx 20\%$) of measurements of out-of-distribution images. Visual comparisons: CS under the Glow prior, DCGAN prior, Lasso-WVT, and Lasso-DCT at a noise level $\sqrt{\mathbb{E}\|\eta\|^2} = 0.1$. In each case, we choose values of the penalization parameter $\gamma$ to yield the best performance. We use $\gamma = 0$ for both DCGAN and Glow priors and $\gamma = 0.01$ for Lasso-WVT, and Lasso-DCT, respectively.

### 3.3 INPAINTING

In inpainting, one is given a masked image of the form $y = M \odot x$, where $M$ is a masking matrix with binary entries and $x \in \mathbb{R}^n$ is an n-pixel image. The goal is to find $x$. We could rewrite (2) with $\gamma = 0$ as

$$\min_{z \in \mathbb{R}^n} \|y - M \odot G(z)\|^2$$

There is an affine space of images consistent with the measurements, and an algorithm must select which is most natural. As before, using the minimizer $\hat{z}$, the estimated image is given by $G(\hat{z})$. Our experiments reveal the same story as for compressed sensing. If initialized at $z_0 = 0$, then the empirical risk formulation with $\gamma = 0$ exhibits high PSNRs on test images. Algorithmic regularization is again occurring due to initialization. In contrast, DCGAN is limited by its representation error. See Figure 6, and Appendix D for more results, including visually reasonable face inpainting, even for out-of-distribution human faces.

Figure 6: Inpainting: Recoveries under DCGAN and Glow, both with $\gamma = 0$.

### 4 DISCUSSION

We have demonstrated that pretrained generative invertible models can be used as natural signal priors in imaging inverse problems. Their strength is that *every desired image is in the range of an invertible model*, and the challenge that they overcome is that *every undesired image is also in the range of the model and no explicit low-dimensional representation is kept*. We study a regularization for empirical loss minimization that promotes recovery of images that have a high value of a proxy for image likelihood

under the generative model. We demonstrate that this formulation can quantitatively and qualitatively outperform BM3D at denoising. Additionally, it has lower recovery errors than Lasso across all levels of undersampling, and it can get comparable errors from 10-20x fewer measurements, which is a 2x reduction from Bora et al. (2017). The superior recovery performance of the invertible prior at very extreme undersampling ratios is particularly surprising given that invertible nets do not maintain explicit low dimensional representations, as GANs do. Additionally, our trained invertible model yields significantly better reconstructions than Lasso even on out-of-distribution images, including images with rare features of variation, and on unrelated natural images.

The idea of analyzing inverse problems with invertible neural networks has appeared in Ardizzone et al. (2018). The authors study estimation of the complete posterior parameter distribution under a forward process, conditioned on observed measurements. Specifically, the authors approximate a particular forward process by training an invertible neural network. The inverse map is then directly available. In order to cope with information loss, the authors augment the measurements with additional variables. This work differs from ours because it involves training a separate net for every particular inverse problem. In contrast, our work studies how to use a pretrained invertible generator for a variety of inverse problems not known at training time. Training invertible networks is challenging and computationally expensive; hence, it is desirable to separate the training of off-the-shelf invertible models from potential applications in a variety of scientific domains.

*Why optimize a proxy for image likelihood instead of optimizing image likelihood directly?*

As noted in Section 2, the immediate formulation one would write down for inverse problems under an invertible prior is to optimize a data misfit term together with an image log-likelihood term. Unfortunately, we found it difficult to get this optimization to converge in practice. The likelihood term can exhibit rapid variation due to the Jacobian of the transformation $z \mapsto x = G(z)$; additionally the likelihood term may in principle even contain local minima or other geometric properties that make gradient descent difficult. Figure 7 compares the loss landscapes in $x$ and $z$, illustrating that the learned likelihood function in $x$ may lead to difficulty in choosing appropriate step sizes for gradient descent algorithms.

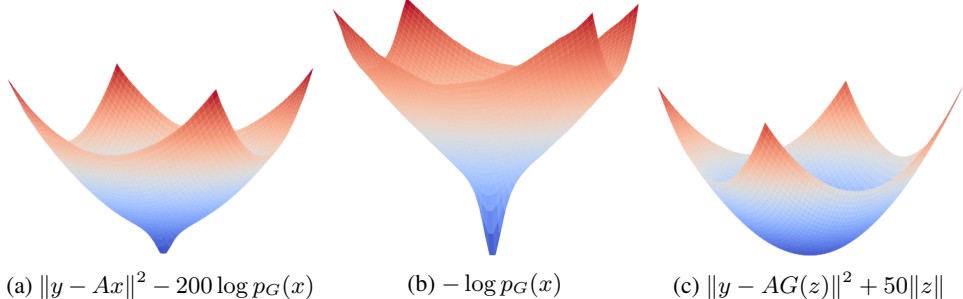

(a) $\|y - Ax\|^2 - 200 \log p_G(x)$  (b) $-\log p_G(x)$  (c) $\|y - AG(z)\|^2 + 50\|z\|$

Figure 7: Landscapes of (a) the loss surface in $x$-space, (b) just the image likelihood in $x$-space, and (c) the loss surface in $z$-space, as functions of two random directions in either $x$ or $z$, as appropriate.

In contrast, there are nice geometric properties that appear in latent space from an invertible model. As an illustration, consider the compressive sensing problem with noiseless measurements. Here, the formulation corresponds to a gradient descent down the data misfit term $\|AG(z) - y\|^2$ starting at $z_0 = 0$. This data misfit term has a favorable geometry for optimization in that all local minima are global minima. This is because the level sets in $z$ of $\|AG(z) - y\|^2$ are given by $G^{-1}$ applied to the level sets in $x$ of $\|Ax - y\|^2$, which have a simple structure because of the linearity of the measurements in $x$. There may be additional benefits due to optimizing in $z$ because the invertible net learns representations that permit interpolation between images and semantically meaningful arithmetic, as reported in Kingma and Dhariwal (2018).

*Why is the likelihood of an image's latent representation a reasonable proxy for the image's likelihood?*

The training process for an invertible generative model attempts to learn a target distribution in images space by directly maximizing the likelihood of provided samples from that distribution, given a standard Gaussian prior in latent space. High probability regions in latent space map to regions in image space of equal probability. Hence, broadly speaking, regions of small values of $\|z\|$ are expected to map to regions of large likelihoods in image space. There will be exceptions to this

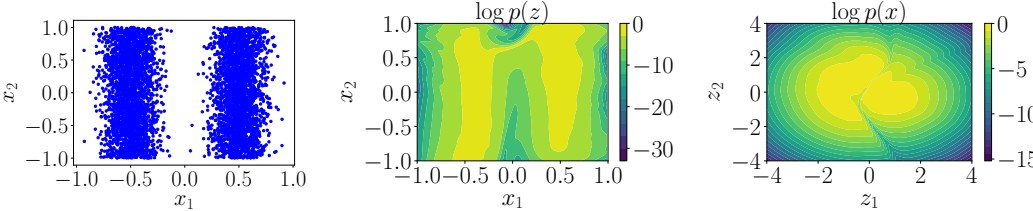

Figure 8: An invertible net was trained on the data points in $x$-space (left), resulting in the given plots of latent $z$-likelihood versus $x$ (middle), and $x$-likelihood versus latent representation $z$ (right).

property. For example, natural image distributions have a multimodal character. The preimage of high probability modes in image space will correspond to high likelihood regions in latent space. Because the generator $G$ is invertible and continuous, interpolation in latent space of these modes will provide images of high likelihood in $z$ but low likelihood in the target distribution. To illustrate this point, we trained a Real-NVP (Dinh et al., 2016) invertible neural network on the two dimensional set of points depicted in Figure 8 (left panel). The middle and right panels show that high likelihood regions in latent space generally correspond to higher likelihood regions in image space, but that there are some regions of high likelihood in latent space that map to points of low likelihood in image space and in the target distribution. We see that the spurious regions are of low total probability and would be unlikely to be the desired outcomes of an inverse problem arising from the target distribution.

*How can solving compressive inverse problems be successful without direct penalization of the proxy image likelihood?*

If there are fewer linear measurements than the dimensionality of the desired signal, an affine space of images is consistent with the measurements. In our formulation, regularization does not occur by direct penalization of our proxy for image likelihood; instead, it occurs implicitly by performing the optimization in $z$-space with an initialization of $z_0 = 0$. The set of latent representations $z$ that are consistent with the compressive measurements define a $m$-dimensional nonlinear manifold. As per the likelihood proxy mentioned above, the spirit of our formulation is to find the point on this manifold that is closest to the origin with respect to the Euclidean norm. Our specific way of estimating this point is to perform a gradient descent down a data misfit term in $z$-space, starting at the origin. While a gradient flow typically will not find the closest point on the manifold, it empirically finds a reasonable approximation of that point. In practice, one could further do a local search to refine the output of this gradient flow, but we elect not to do so for the sake of simplicity.

*Why does the invertible prior do so well, especially on out-of-distribution images?*

One reason that the invertible prior performs so well is because it has no representation error. The lack of representation error of invertible nets presents a significant opportunity for imaging with a learned prior. Any image is potentially recoverable, even if the image is significantly outside of the training distribution. In contrast, methods based on projecting onto an explicit low-dimensional representation of a natural signal manifold will have representation error, perhaps due to modeling assumptions, mode collapse, or bias in a training set. Such methods will see performance prematurely saturate as the number of measurements increases. In contrast, an invertible prior would not see performance saturate. In the extreme case of having a full set of exact measurements, an invertible prior could in principle recover any image exactly.

It is natural to wonder which images can be effectively recovered using an invertible prior trained on a particular signal class. As expected, we see the best reconstruction errors on in-distribution images and performance degrades as images get further out-of-distribution. Nonetheless, we observe that reconstruction errors of unrelated natural images are still of higher quality than with the Lasso. It appears that the invertible generator learns some general attributes of natural images. This leads to several questions: when a generative invertible net is trained, how far out-of-distribution can an image be while maintaining a high likelihood? How do invertible nets learn useful statistics of natural images? Is that due primarily to training, or is there architectural bias toward natural images, as with the Deep Image Prior and Deep Decoder (Ulyanov et al., 2018; Heckel and Hand, 2018)?

The results of this paper provide further evidence that reducing representational error of generators can significantly enhance the performance of generative models for inverse problems in imaging. This idea was also recently explored in Athar et al. (2018), where the authors trained a GAN-like prior with a high-dimensional latent space. The high dimensionality of this space lowers representational error, though it is not zero. In their work, the high-dimensional latent space had a structure that was difficult to directly optimize, so the authors successfully modeled latent representations as the output of an untrained convolutional neural network whose parameters are estimated at test time. Their paper and ours raises several questions: Which generator architectures provide a good balance between low representation error, ease of training, and ease of inversion? Should a generative model be capable of producing all images in order to perform well on out-of-distribution images of interest? Are there cheaper architectures that perform comparably? These questions are quite important, as solving equation 2 in our $64 \times 64$ pixel color images experiments took 15 GPU-minutes. New developments are needed on architectures and frameworks in between low-dimensional generative priors and fully invertible generative priors. Such methods could leverage the strengths of invertible models while being much cheaper to train and use.

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

## A    EXPERIMENTAL SETUP

Simulations were completed mainly on CelebA-HQ dataset, used in Kingma and Dhariwal (2018); it has 30,000 color images that were resized to $64 \times 64$ for computational reasons, and were split into 27,000 training and 3000 test images. We also provide some additional experiments on the Flowers Nilsback and Zisserman (2008), and Birds Wah et al. (2011) datasets. Flowers dataset contains 8189 color images resized to $64 \times 64$ out of which $500$ images are spared for testing. Birds dataset contains a total of 11,788 images, which were center aligned and resized to $64 \times 64$ out of which $5794$ images are set aside for testing.

We specifically model our invertible networks after the recently proposed Glow Kingma and Dhariwal (2018) architecture, which consists of a multiple flow steps. Each flow step comprises of an activation normalization layer, a $1 \times 1$ convolutional layer, and an affine coupling layer, each of which is invertible. Let $K$ be the number of steps of flow before a splitting layer, and $L$ be the number of times the splitting is performed. To train over CelebA, we choose the network to have $K = 48$, $L = 4$ and affine coupling, and train it with a learning rate $0.0001$, and a batch size 6 at resolution $64 \times 64 \times 3$. The model was trained over $5-$bit images with 10,000 warmup iterations as in Kingma and Dhariwal (2018), but when solving inverse problems using Glow original $8-$bit images were used. We refer the reader to Kingma and Dhariwal (2018) for specific details on the operations performed in each of the network layer.

We use LBFGS to solve the inverse problem. For best performance, we set the number of iterations and learning rate for denoising, compressed sensing, and inpainting to be 20, 1; 30, 0.1; and 20, 1; respectively. we use Pytorch to implement Glow network training and solve the inverse problem. Glow training was conducted on a single Titan Xp GPU using a maximum allowable (under given computational constraints) batch size of 6. In case of CS, recovering a single image on Titan Xp using LBFGS solver with 30 steps takes $889.125$ seconds ($14.82$ minutes). However, we can solve 6 inverse problems in parallel on the given hardware platform.

Unless specified otherwise, inverse problem under Glow prior is always initialized with $z_0 = 0$. Whereas under DCGAN prior, we initialize with $z_0 \sim \mathcal{N}(0, 0.1^2 I)$ and report average over three random restarts. In all the quantitative experiments over, the reported quality metrics such as PSNR, and reconstruction errors are averaged over 12 randomly drawn test set images.

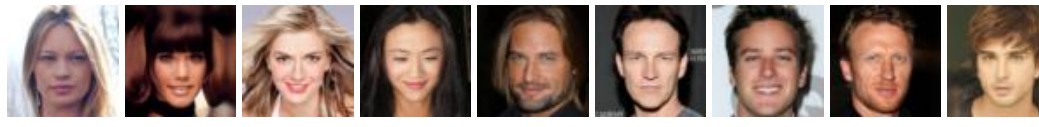

Figure 9: Samples from training set of CelebA downsampled to $64 \times 64 \times 3$.

## B    DENOISING: ADDITIONAL EXPERIMENTS

We present additional quantitative experiments on image denoising here. Complete set of experiments on average PSNR over 12 CelebA (within distribution[2]) test set images versus penalization parameter $\gamma$ under noise levels $\sigma = 0.01, 0.05, 0.1$, and $0.2$ are presented in Figure 10 below. The central message is that Glow prior outperforms DCGAN prior uniformly across all $\gamma$ due to the representation limit of DCGAN. In addition, striking the right balance between the misfit term and the penalization term by appropriately choosing $\gamma$ improves the performance of Glow, and it also approaches state-of-the-art BM3D algorithm at low noise levels, and clearly visible in higher noise, for example, at a noise level of $\sigma = 0.2$, the Glow prior improves upon BM3D by 2dB. Visually the results of Glow prior are clearly even superior to BM3D recoveries that are generally blurry and over smoothed as can be spotted in the qualitative results below. To avoid fitting the noisy image using the Glow model, we force the recoveries to be natural by choosing large enough $\gamma$.

---

[2]The redundant 'within distribution' phrase is added to emphasize that the test set images are drawn from the same distribution as the train set. We do this to avoid confusion with the out-of-distribution recoveries also presented in this paper.

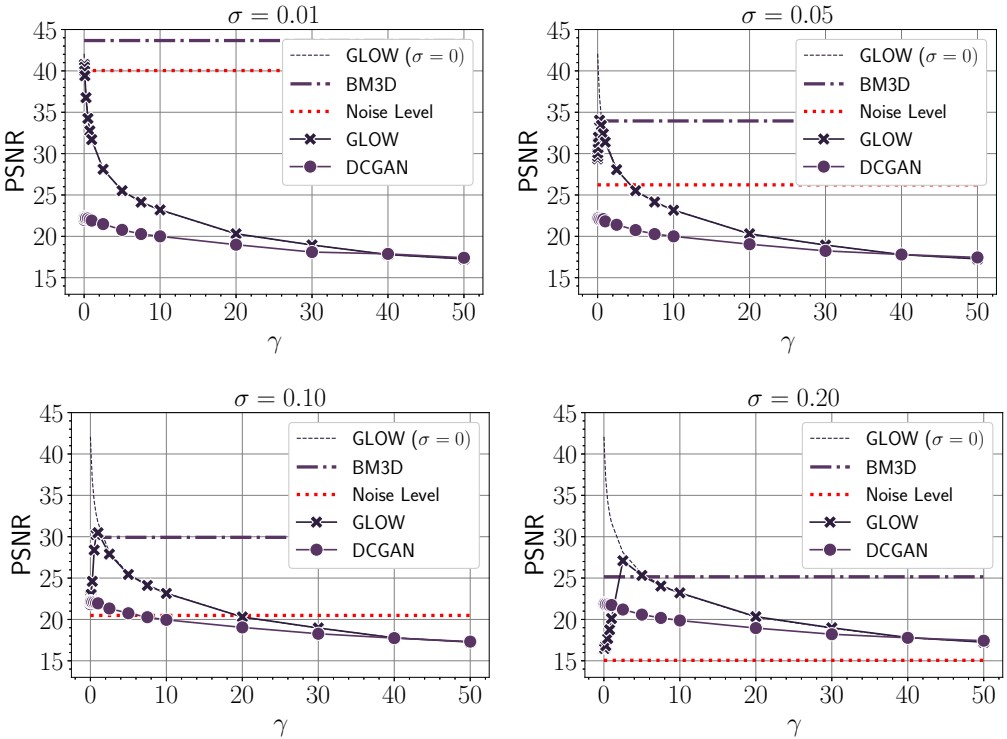

Figure 10: Image Denoising — Recovered PSNR values as a function of $\gamma$ under Glow prior, and DCGAN prior on (within-distribution) test set CelebA images. For reference, we show the average PSNRs of the original noisy images, and under the Glow prior in the noiseless case ($\sigma = 0$) in both panels. The average PSNR after applying BM3D, and the average PSNR under the Glow prior at noise levels $\sigma = 0.01, 0.05, 0.10, 0.20$ are reported.

Recall that we are solving a regularized empirical risk minimization program

$$\arg\min_{z \in \text{Domain}(G)} \|y - AG(z)\|^2 + \gamma\|z\|.$$

In general, one can instead solve $\arg\min_{z \in \text{Domain}(G)} \|y - AG(z)\|^2 + H(\|z\|)$, where $H(\cdot)$ is a monotonically increasing function. Figure 11 shows the comparison of most common choices of linear (already used in the rest of the paper), and quadratic $H$ in the context of densoing. We find that the highest achievable PSNR remains the same in both the cases, however, the penalization parameter $\gamma$ has to be adjusted accordingly.

We train Glow and DCGAN on CelebA. Additional qualitative image denosing results under higher noise level $\sigma = 0.1$ and $0.2$ comparing Glow prior against DCGAN prior, and BM3D are presented below in Figure 12, and 13.

We also trained Glow model on Flowers dataset. Below we present its qualitative denoising performance against BM3D on the test set Flowers images. We also show the effect of varying $\gamma$ — smaller $\gamma$ leads to overfitting and vice versa.

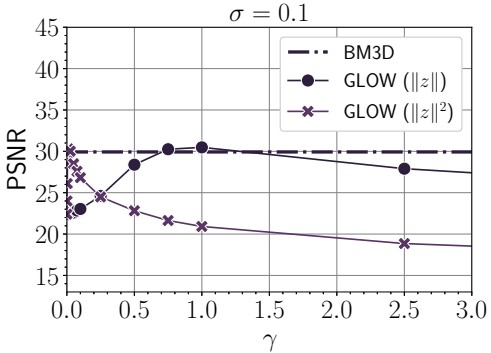

Figure 11: Image Denoising — Recovered PSNR values as a function of $\gamma$ under Glow prior with $\|z\|$ and $\|z\|^2$ penalization on (within-distribution) test set CelebA images. Comparison is provided with BM3D denoising at noise level $\sigma = 0.1$

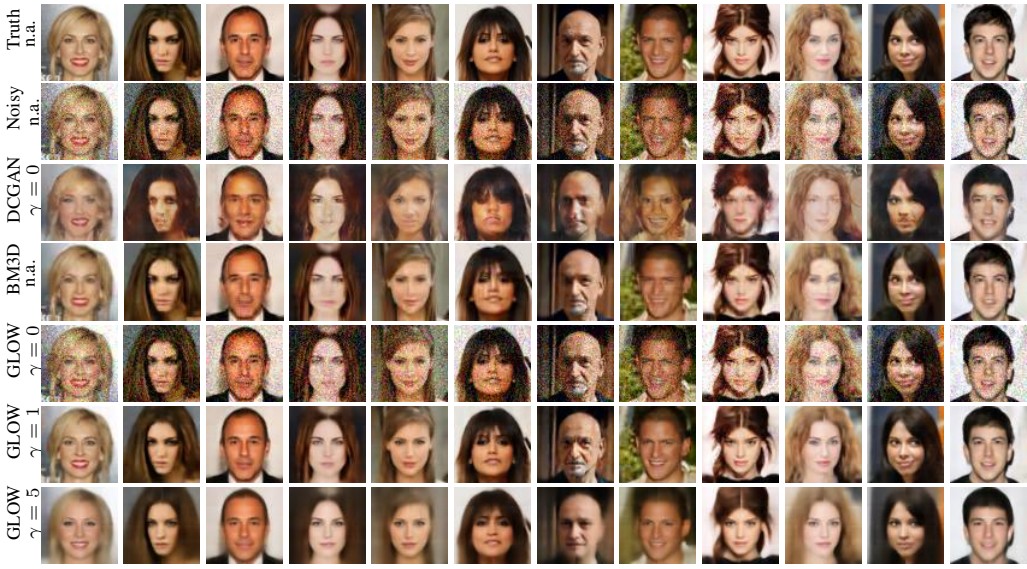

Figure 12: Image Denoising — Visual comparisons under the Glow prior, the DCGAN prior, and BM3D at a noise level $\sigma = 0.1$ on CelebA (within-distribution) test set images. Under DCGAN prior, we only show the case of $\gamma = 0$ as this consistently gave the best performance for DCGAN. Under Glow prior, the best performance over is achieved with $\gamma = 1$, overfitting of the image occurs with $\gamma = 0$ and underfitting occurs at $\gamma = 5$. Note that the Glow prior with $\gamma = 1$ also gives a sharper image than BM3D.

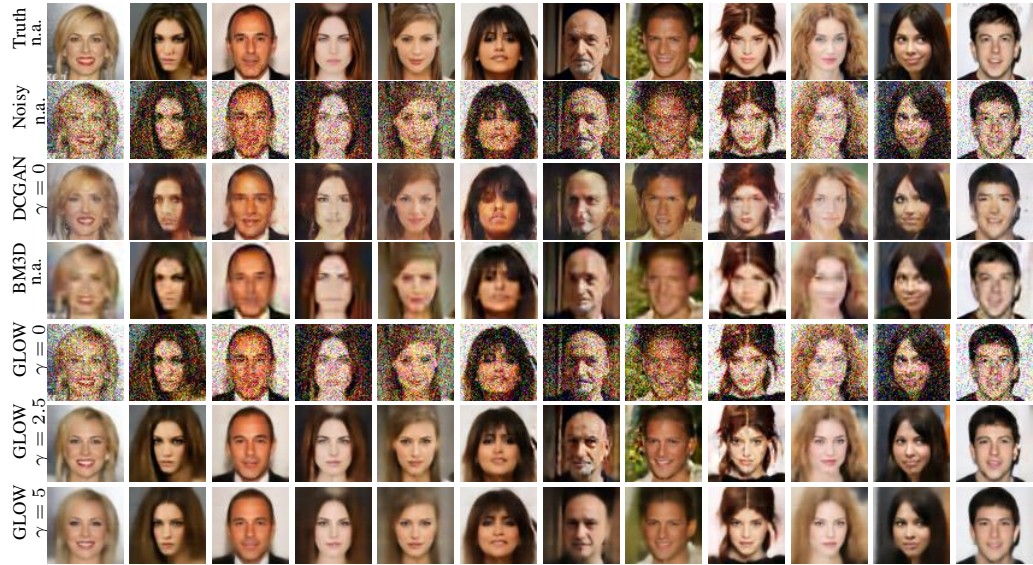

Figure 13: Image Denoising — Visual comparisons under the Glow prior, the DCGAN prior, and BM3D at noise level $\sigma = 0.2$ on CelebA (within-distribution) test set images. Under DCGAN prior, we only show the case of $\gamma = 0$ as this consistently gives the best performance. Under Glow prior, the best performance is achieved with $\gamma = 2.5$, overfitting of the image occurs with $\gamma = 0$ and underfitting occurs with $\gamma = 5$. Note that the Glow prior with $\gamma = 2.5$ also gives a sharper image than BM3D.

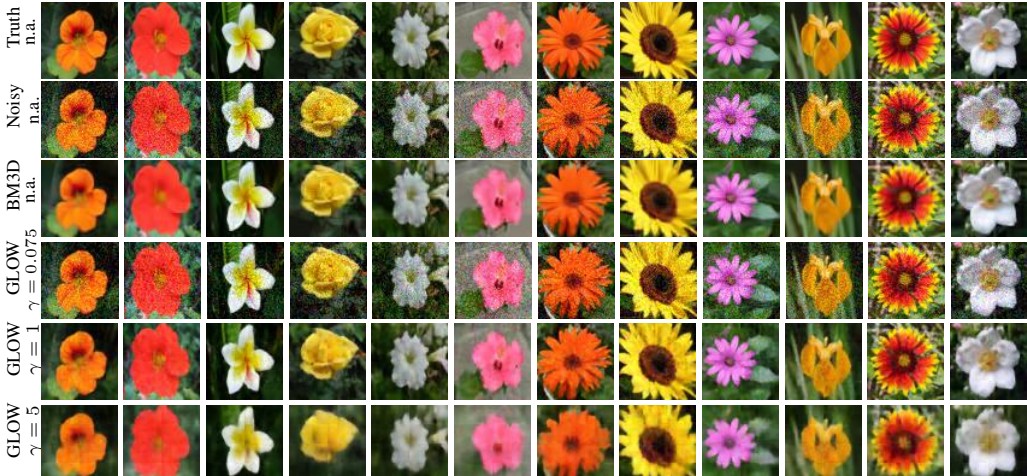

Figure 14: Image Denoising — Visual comparisons under the Glow prior, and BM3D at noise level $\sigma = 0.1$ on (within-distribution) test set Flowers images. Under Glow prior, the best performance is obtained with $\gamma = 1$. Note that the Glow prior with $\gamma = 1$ also gives a sharper image than BM3D.

## C    COMPRESSED SENISNG: ADDITIONAL EXPERIMENTS

Some additional quantitative image recovery results on test set of CelebA dataset are presented in Figure 15; it depicts the comparison of Glow prior, DCGAN prior, LASSO-DCT, and LASSO-WVT at compressed sensing. We plot the reconstruction error : $= \frac{1}{n}\|x - \hat{x}\|_2^2$, where $\hat{x}$ is the recovered image and $n = 12288$ is the number of pixels in the $64 \times 64 \times 3$ CelebA images. Glow uniformly outperforms DCGAN, and LASSO across entire range of the number of measuremnts. LASSO-DCT and LASSO-WVT eventually catch up to Glow but only when observed measurements are a significant fraction of the total number of pixels. On the other hand, DCGAN is initially better than LASSO but prematurely saturates due to limited representation capacity.

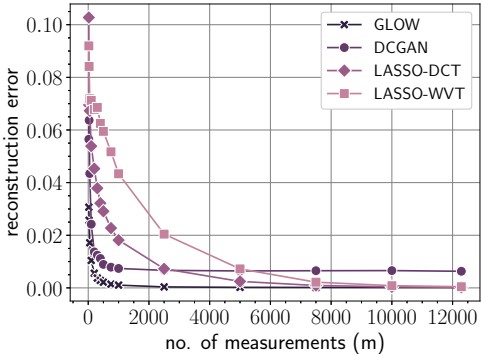

Figure 15: Compressed sensing — Reconstruction error vs. number of measurements under Glow prior, DCGAN prior, LASSO-DCT and LASSO-WVT on CelebA (within-distribution) test set images. Noise $\eta$ is scaled such that $\mathbb{E}\|\eta\|^2 = 0.01$ and the penalization parameter $\gamma = 0$ for Glow, and DCGAN; and $\gamma = 0.01$ for LASSO-DCT, and LASSO-WVT.

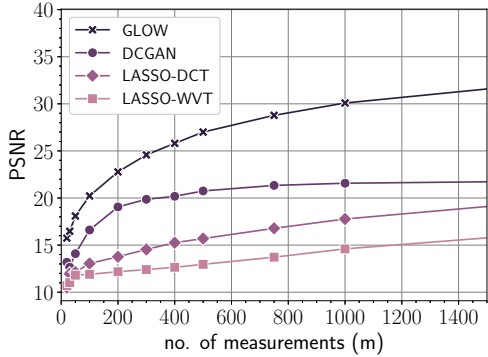

Figure 16: Compressed sensing — Zoomed-in version of the left panel of Figure 4 in the main paper in the low measurement regime for CelebA. PSNR vs. number of measurements under Glow prior, DCGAN prior, LASSO-DCT and LASSO-WVT on the CelebA (within distribution) test set images. Noise $\eta$ is scaled such that $\sqrt{\mathbb{E}\|\eta\|^2} = 0.1$ and the penalization parameter $\gamma = 0$ for Glow and DCGAN; and $\gamma = 0.01$ for LASSO-DCT, and LASSO-WVT.

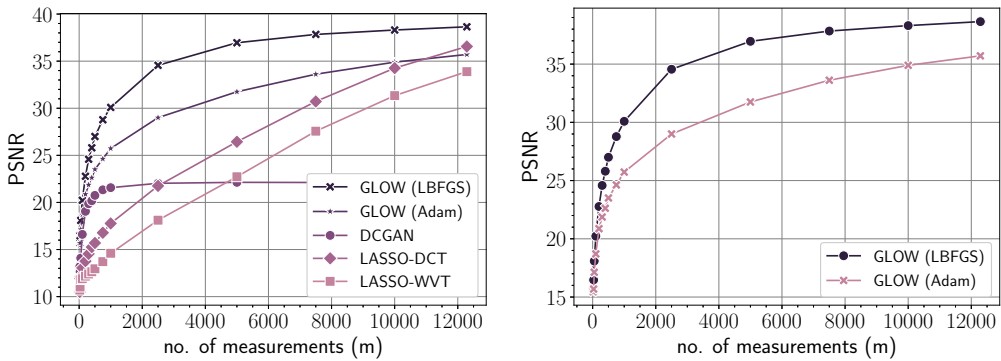

Figure 17: Compressed sensing under Glow prior. Performance comparison between LBFGS and Adam solver for the inverse problem. For Adam solver, 2000 gradient steps were taken with learning rate chosen to be 0.01. The rest of the parameters were fixed to be the same as with LBFGS.

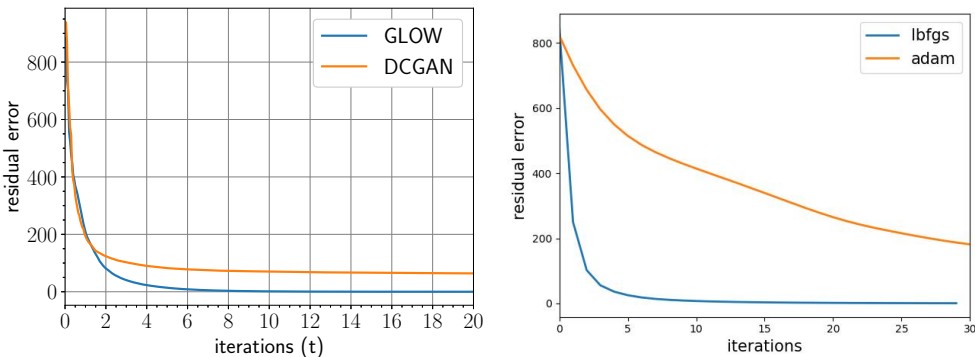

Figure 18: Residual error vs. number of iterations. Left panel compares DCGAN and Glow priors. Both converge roughly at the same rate to their respective saturation levels. The right panel compares LBFGS and Adam solvers for compressed sensing under Glow prior. LBFGS tends to converge far more quickly than Adam. We choose $\gamma = 0$ in both the experiments.

Surprisingly, we observe that no explicit penalization of likelihood is necessary for compressive sensing with an invertible generative prior under formulation equation 2. That is, we can take $\gamma = 0$ when the optimization is initialized at $z_0 = 0$. This indicates that algorithmic regularization is occurring and that initialization plays a role. We performed some additional experiments to study the role of initialization. The left panel in Figure 19 shows that as the norm of the latent initialization increases, the norm of the recovered latent representation increases and the PSNR of the recovered image decreases. Moreover, the right panel in Figure 19 shows the norm of the estimated latent representation at each iteration of the optimization. In all our experiments, it monotonically grows versus iteration number. These experiments provide further evidence that smaller latent initializations lead to outputs that are more natural and have smaller latent representations.

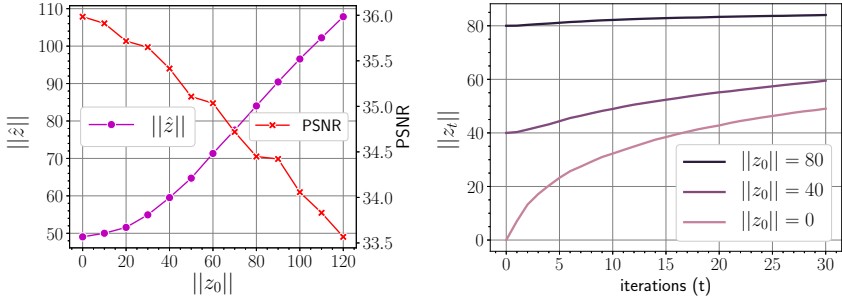

Figure 19: The left panel shows the average PSNR over 12 test set images and norm of the optimizer $\hat{z}$ as a function of the norm of the initialization for the LBFGS solver to equation 2 for Compressed sensing under Glow prior with $\gamma = 0$. The initialization $z_0$ was chosen randomly and rescaled to the desired norm. The right panel shows the norm of the estimated latent representation as a function of iteration number for multiple initializations. The Adam solver behaves similarly.

Recall that the natural face images correspond to smaller $z_0$. In Figure 20, we plot the norm of the latent codes of the iterates of each algorithm vs. the number of iterations. The central message is that initializing with smaller norm $z_0$ tends to yield natural (smaller latent representations) recoveries. This is one explanation as to why in compressed sensing, one is able to obtain the true solution out of the affine space of solutions without penalizing the unnaturalness of the recoveries.

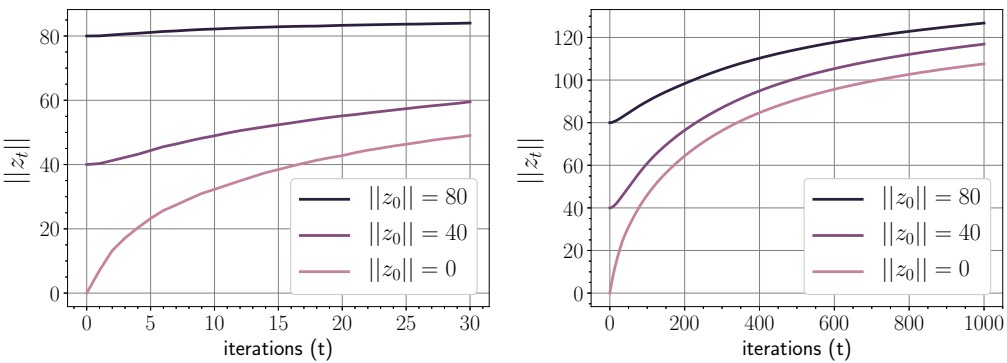

Figure 20: Compressed sensing — Norm of the latent codes with iterations. Left panel shows how the norm of the latent codes evolves over iterations of the LBFGS solver under different size initializations. Right panel shows the same experiment for the Adam solver (although over much larger number of iterations as Adam requires comparatively more iterations to converge). Each point is averaged over 12 test set images under random rescaled initializations $z_0$. We set the penalization parameter $\gamma = 0$ in both experiments.

We now present visual recovery results on test images from the CelebA dataset under varying number of measurements in compressed sesing. We compare recoveries under Glow prior, DCGAN prior, LASSO-DCT, and LASSO-WVT.

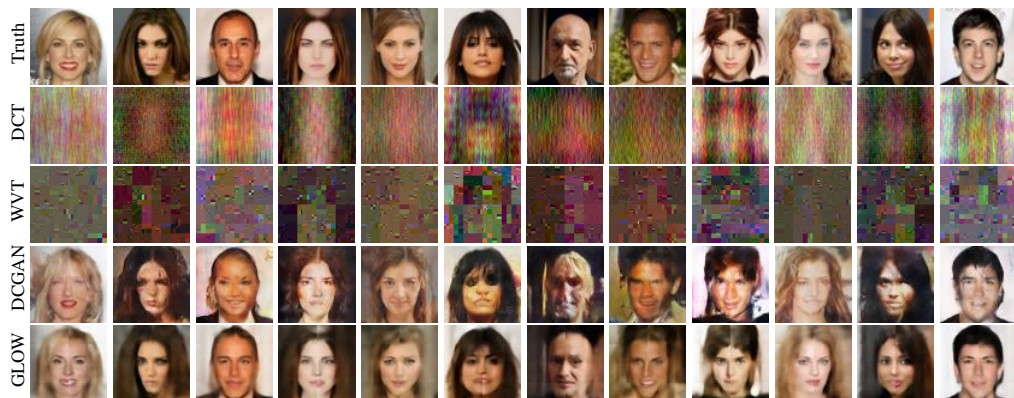

Figure 21: Compressed sensing visual comparisons — Recoveries on (within-distribution) test set images with a number $m = 200$ ($\approx 1.5\%$) of measurements under the Glow prior, the DCGAN prior, LASSO-WVT, and LASSO-DCT at a noise level $\sqrt{\mathbb{E}\|\eta\|^2} = 0.1$. In each case, we choose values of the penalization parameter $\gamma$ to yield the best performance among the tested values. We use $\gamma = 0$ for both DCGAN, and Glow prior and $\gamma = 0.01$ for LASSO-WVT, and LASSO-DCT, respectively.

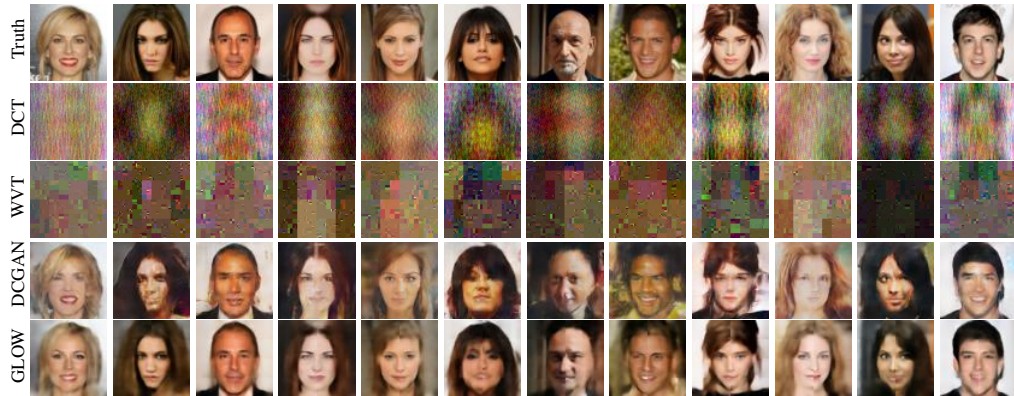

Figure 22: Compressed sensing visual comparisons — Recoveries on the (within-distribution) test set images with a number $m = 300$ ($\approx 2\%$) of measurements under the Glow prior, the DCGAN prior, LASSO-WVT, and LASSO-DCT at a noise level $\sqrt{\mathbb{E}\|\eta\|^2} = 0.1$. In each case, we choose values of the penalization parameter $\gamma$ to yield the best performance among the tested values. We use $\gamma = 0$ for both DCGAN, and Glow prior and $\gamma = 0.01$ for LASSO-WVT, and LASSO-DCT, respectively.

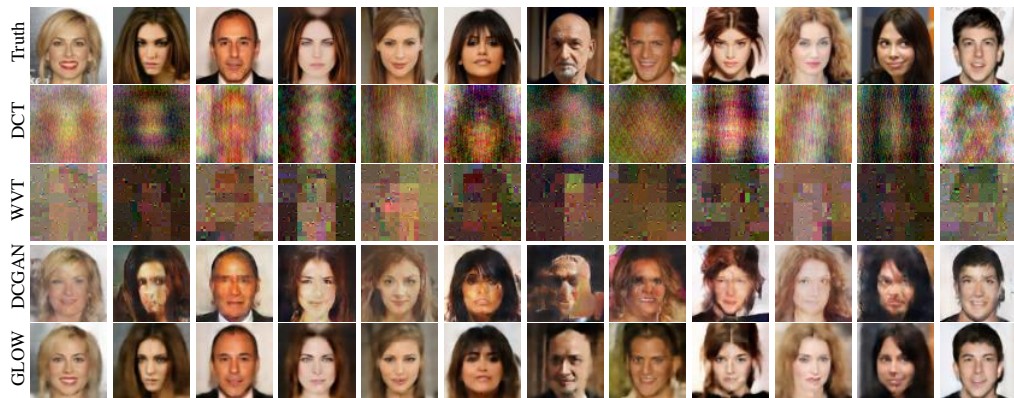

Figure 23: Compressed sensing visual comparisons — Recoveries on (within-distribution) test set images with a number $m = 400$ ($\approx 3\%$) of measurements under the Glow prior, the DCGAN prior, LASSO-WVT, and LASSO-DCT at a noise level $\sqrt{\mathbb{E}\|\eta\|^2} = 0.1$. In each case, we choose values of the penalization parameter $\gamma$ to yield the best performance among the tested values. We use $\gamma = 0$ for both DCGAN, and Glow prior and $\gamma = 0.01$ for LASSO-WVT, and LASSO-DCT, respectively.

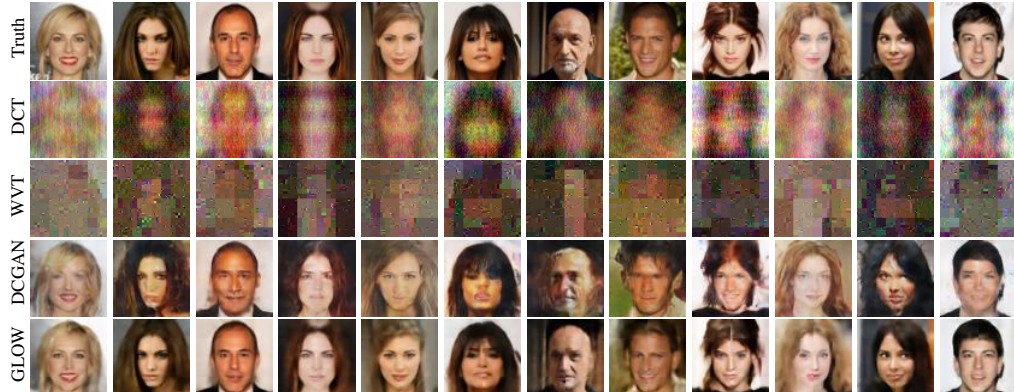

Figure 24: Compressed sensing visual comparisons — Recoveries on (within-distribution) test set images with a number $m = 500$ ($\approx 4\%$) of measurements under the Glow prior, the DCGAN prior, LASSO-WVT, and LASSO-DCT at a noise level $\sqrt{\mathbb{E}\|\eta\|^2} = 0.1$. In each case, we choose values of the penalization parameter $\gamma$ to yield the best performance among the tested values. We use $\gamma = 0$ for both DCGAN, and Glow prior and $\gamma = 0.01$ for LASSO-WVT, and LASSO-DCT, respectively.

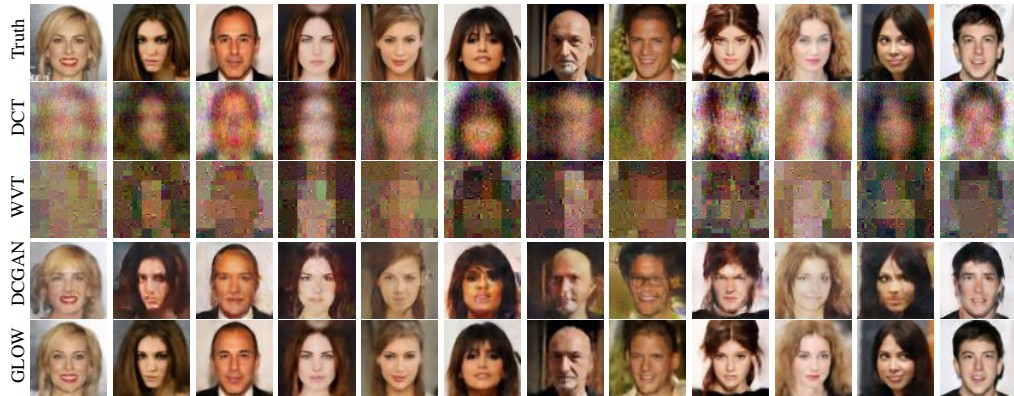

Figure 25: Compressed sensing visual comparisons — Recoveries on (within-distribution) test set images with a number $m = 750$ ($\approx 6\%$) of measurements under the Glow prior, the DCGAN prior, LASSO-WVT, and LASSO-DCT at a noise level $\sqrt{\mathbb{E}\|\eta\|^2} = 0.1$. In each case, we choose values of the penalization parameter $\gamma$ to yield the best performance among the tested values. We use $\gamma = 0$ for both DCGAN, and Glow prior and $\gamma = 0.01$ for LASSO-WVT, and LASSO-DCT, respectively.

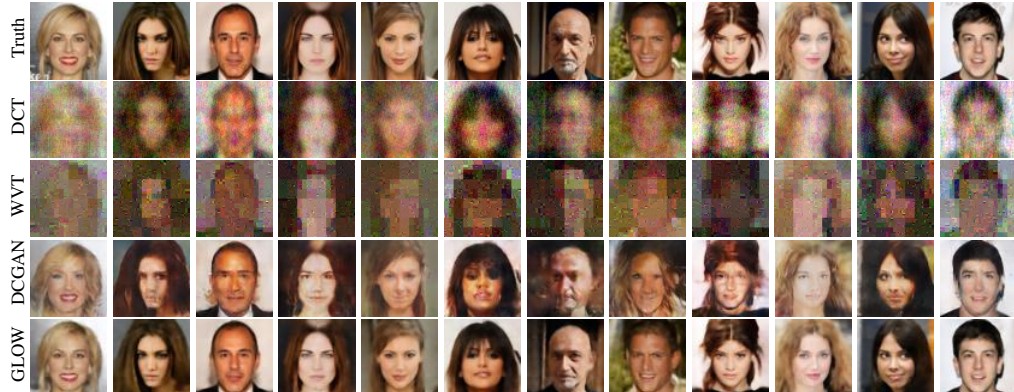

Figure 26: Compressed sensing visual comparisons — Recoveries on (within-distribution) test set images with a number $m = 1000$ ($\approx 8\%$) of measurements under the Glow prior, the DCGAN prior, LASSO-WVT, and LASSO-DCT at a noise level $\sqrt{\mathbb{E}\|\eta\|^2} = 0.1$. In each case, we choose values of the penalization parameter $\gamma$ to yield the best performance among the tested values. We use $\gamma = 0$ for both DCGAN, and Glow prior and $\gamma = 0.01$ for LASSO-WVT, and LASSO-DCT, respectively.

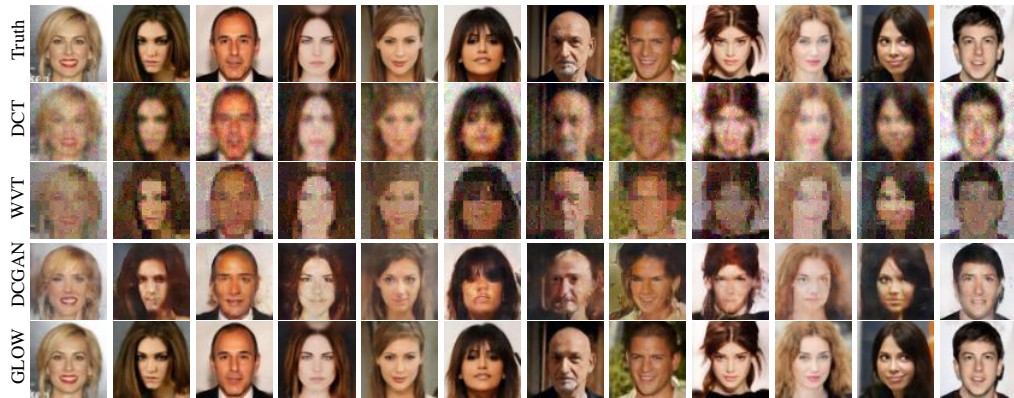

Figure 27: Compressed sensing visual comparisons — Recoveries on (within-distribution) test set images with a number $m = 2500$ ($\approx 20\%$) of measurements under the Glow prior, the DCGAN prior, LASSO-WVT, and LASSO-DCT at a noise level $\sqrt{\mathbb{E}\|\eta\|^2} = 0.1$. In each case, we choose values of the penalization parameter $\gamma$ to yield the best performance among the tested values. We use $\gamma = 0$ for both DCGAN, and Glow prior and $\gamma = 0.01$ for LASSO-WVT, and LASSO-DCT, respectively.

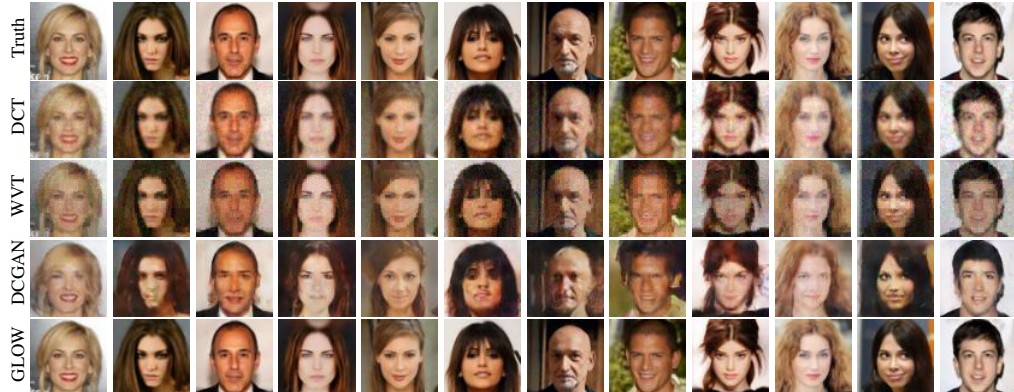

Figure 28: Compressed sensing visual comparisons — Recoveries on (within-distribution) test set images with a number $m = 5000$ ($\approx 41\%$) of measurements under the Glow prior, the DCGAN prior, LASSO-WVT, and LASSO-DCT at a noise level $\sqrt{\mathbb{E}\|\eta\|^2} = 0.1$. In each case, we choose values of the penalization parameter $\gamma$ to yield the best performance among the tested values. We use $\gamma = 0$ for both DCGAN, and Glow prior and $\gamma = 0.01$ for LASSO-WVT, and LASSO-DCT, respectively.

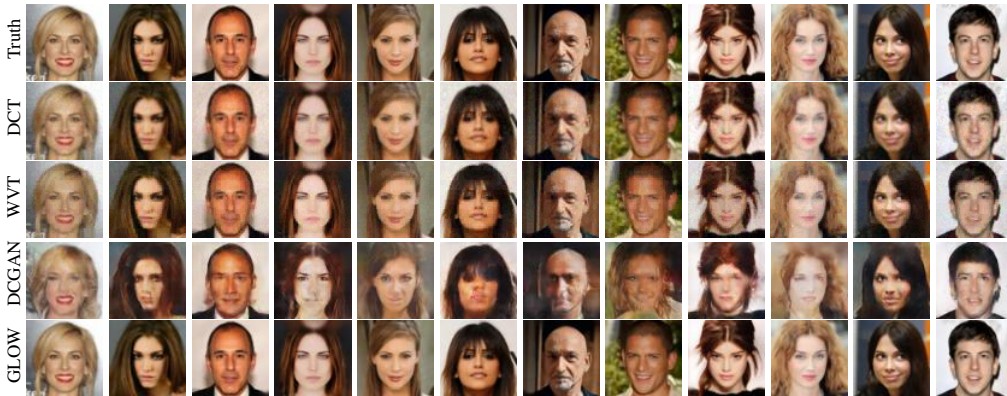

Figure 29: Compressed sensing visual comparisons — Recoveries on (within-distribution) test set images with a number $m = 7500 (\approx 61\%)$ of measurements under the Glow prior, the DCGAN prior, LASSO-WVT, and LASSO-DCT at a noise level $\sqrt{\mathbb{E}\|\eta\|^2} = 0.1$. In each case, we choose values of the penalization parameter $\gamma$ to yield the best performance among the tested values. We use $\gamma = 0$ for both DCGAN, and Glow prior and $\gamma = 0.01$ for LASSO-WVT, and LASSO-DCT, respectively.

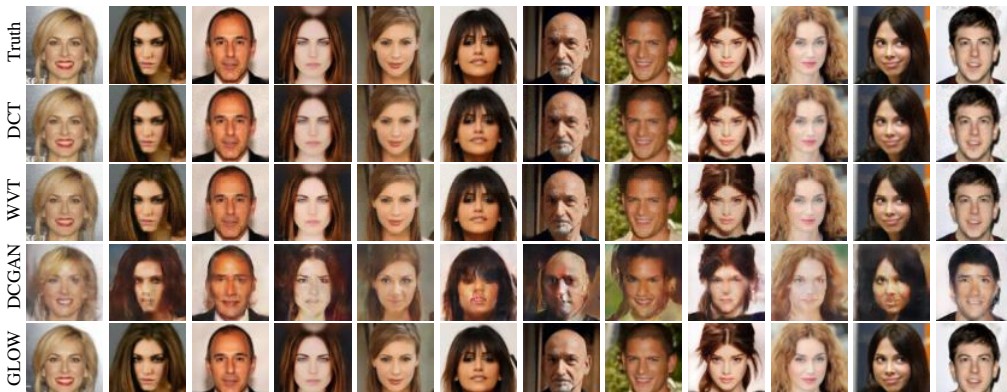

Figure 30: Compressed sensing visual comparisons — Recoveries on (within-distribution) test set images with a number $m = 10,000 (\approx 81\%)$ of measurements under the Glow prior, the DCGAN prior, LASSO-WVT, and LASSO-DCT at a noise level $\sqrt{\mathbb{E}\|\eta\|^2} = 0.1$. In each case, we choose values of the penalization parameter $\gamma$ to yield the best performance among the tested values. We use $\gamma = 0$ for both DCGAN, and Glow prior and $\gamma = 0.01$ for LASSO-WVT, and LASSO-DCT, respectively.

### C.1 COMPRESSED SENSING ON FLOWER AND BIRD DATASET

We also performed compressed sensing experiments similar to those on CelebA dataset above on Birds dataset, and Flowers dataset. We trained a Glow invertible network for each dataset, and present below the quantitative and qualitative recoveries for each dataset.

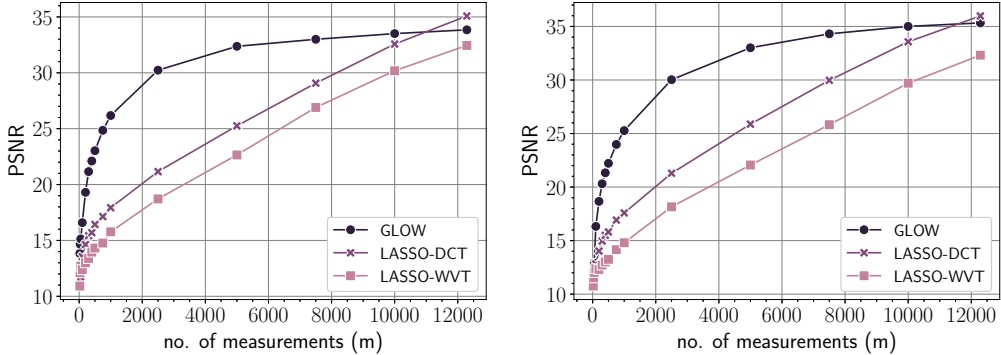

Figure 31: PSNR vs. number of measurements $m$ in compressed sensing under Glow prior, LASSO-DCT and LASSO-WVT on Birds dataset (left panel) and Flowers dataset (right panel). Noise $\eta$ is scaled such that $\sqrt{\mathbb{E}\|\eta\|^2} = 0.1$ and the penalization parameter $\gamma = 0$ for Glow, and $\gamma = 0.01$ for LASSO-DCT, and LASSO-WVT.

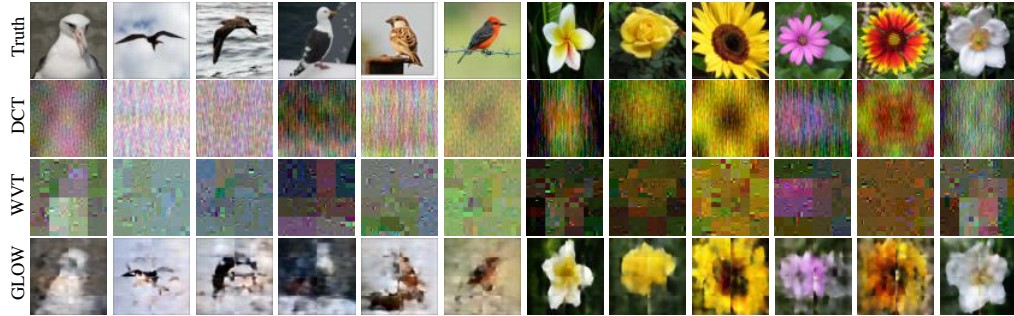

Figure 32: Compressed sensing — Visual comparisons on (within-distribution) test set images from Birds and Flowers dataset with a number $m = 200$ ($\approx 1.5\%$) of measurements under the Glow prior, LASSO-WVT, and LASSO-DCT at a noise level $\sqrt{\mathbb{E}\|\eta\|^2} = 0.1$. In each case, we choose values of the penalization parameter $\gamma$ to yield the best performance among the tested values. We use $\gamma = 0$ for Glow prior and $\gamma = 0.01$ for LASSO-WVT, and LASSO-DCT, respectively.

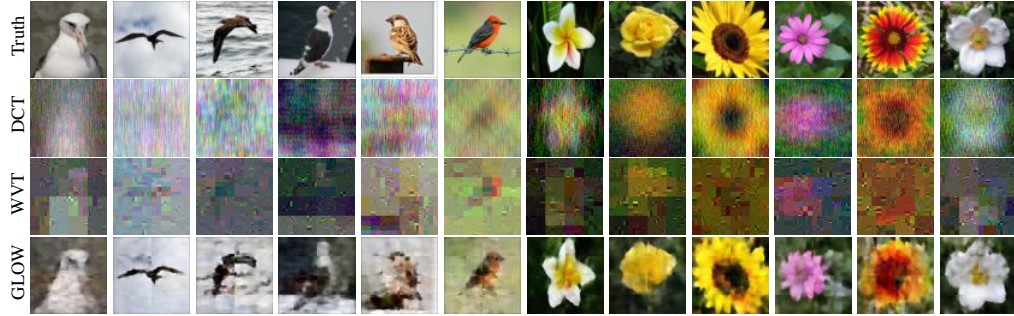

Figure 33: Compressed sensing — Visual comparisons on (within-distribution) test set images from Birds and Flowers dataset with a number $m = 300 \, (\approx 2\%)$ of measurements under the Glow prior, LASSO-WVT, and LASSO-DCT at a noise level $\sqrt{\mathbb{E}\|\eta\|^2} = 0.1$. In each case, we choose values of the penalization parameter $\gamma$ to yield the best performance among the tested values. We use $\gamma = 0$ for Glow prior and $\gamma = 0.01$ for LASSO-WVT, and LASSO-DCT, respectively.

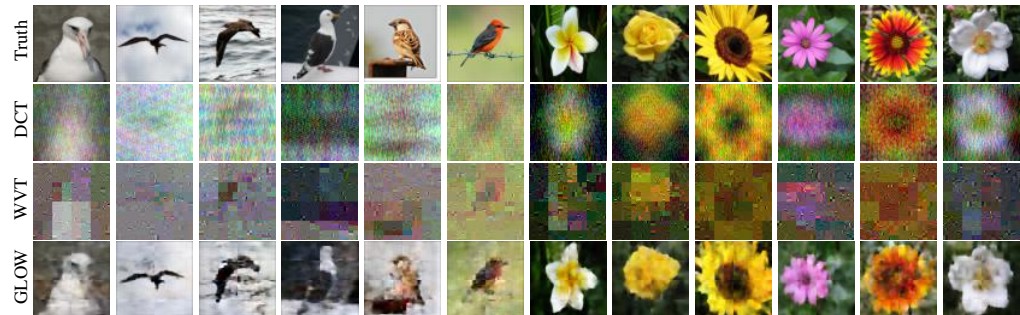

Figure 34: Compressed sensing — Visual comparisons on (within-distribution) test set images from Birds and Flowers dataset with a number $m = 400 \, (\approx 3\%)$ of measurements under the Glow prior, LASSO-WVT, and LASSO-DCT at a noise level $\sqrt{\mathbb{E}\|\eta\|^2} = 0.1$. In each case, we choose values of the penalization parameter $\gamma$ to yield the best performance among the tested values. We use $\gamma = 0$ for Glow prior and $\gamma = 0.01$ for LASSO-WVT, and LASSO-DCT, respectively.

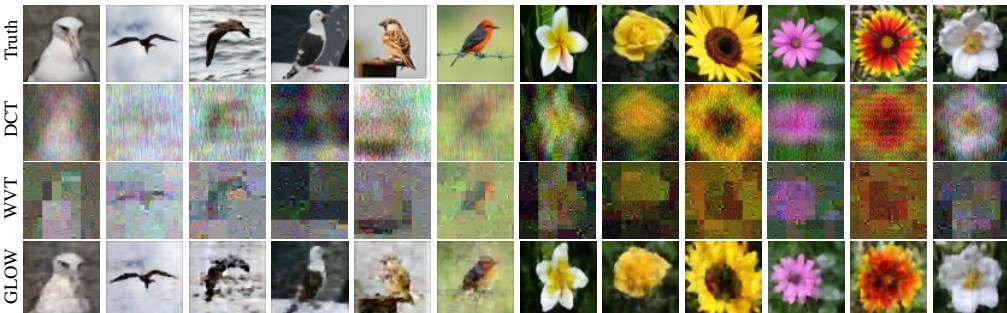

Figure 35: Compressed sensing — Visual comparisons on (within-distribution) test set images from Birds and Flowers dataset with a number $m = 500 \, (\approx 4\%)$ of measurements under the Glow prior, LASSO-WVT, and LASSO-DCT at a noise level $\sqrt{\mathbb{E}\|\eta\|^2} = 0.1$. In each case, we choose values of the penalization parameter $\gamma$ to yield the best performance among the tested values. We use $\gamma = 0$ for Glow prior and $\gamma = 0.01$ for LASSO-WVT, and LASSO-DCT, respectively.

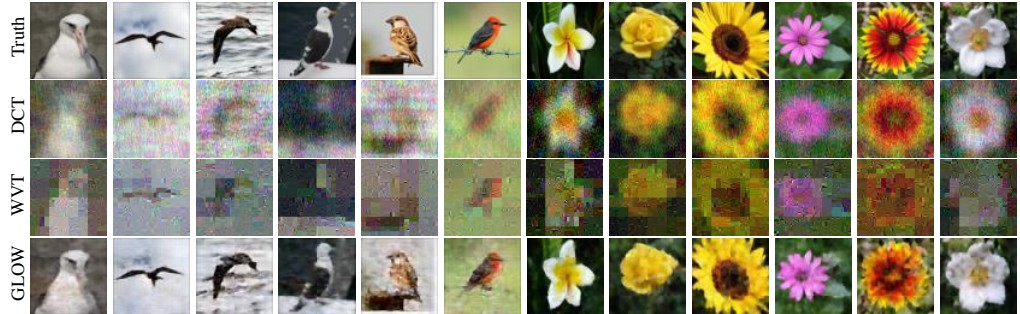

Figure 36: Compressed sensing — Visual comparisons on the test set images from Birds and Flowers dataset with a number $m = 750$ ($\approx 6\%$) of measurements under the Glow prior, LASSO-WVT, and LASSO-DCT at a noise level $\sqrt{\mathbb{E}\|\eta\|^2} = 0.1$. In each case, we choose values of the penalization parameter $\gamma$ to yield the best performance among the tested values. We use $\gamma = 0$ for Glow prior and $\gamma = 0.01$ for LASSO-WVT, and LASSO-DCT, respectively.

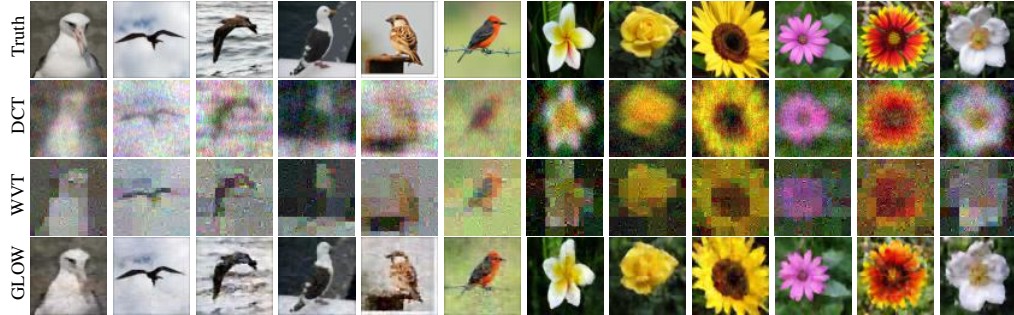

Figure 37: Compressed sensing — Visual comparisons on the test set images from Birds and Flowers dataset with a number $m = 1,000$ ($\approx 8\%$) of measurements under the Glow prior, LASSO-WVT, and LASSO-DCT at a noise level $\sqrt{\mathbb{E}\|\eta\|^2} = 0.1$. In each case, we choose values of the penalization parameter $\gamma$ to yield the best performance among the tested values. We use $\gamma = 0$ for Glow prior and $\gamma = 0.01$ for LASSO-WVT, and LASSO-DCT, respectively.

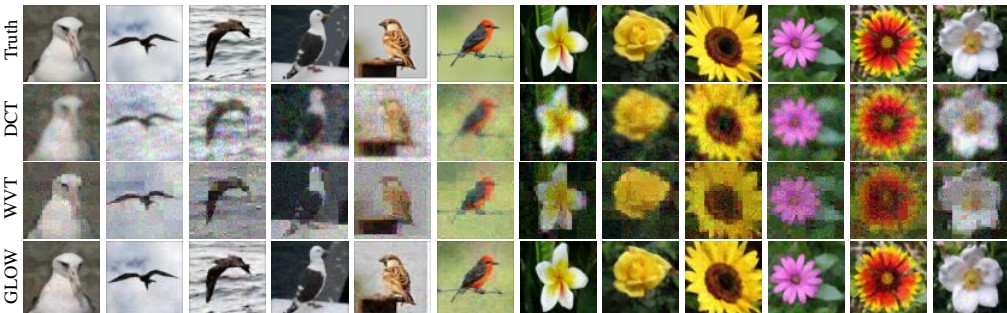

Figure 38: Compressed sensing — Visual comparisons on the test set images from Birds and Flowers dataset with a number $m = 2,500$ ($\approx 20\%$) of measurements under the Glow prior, LASSO-WVT, and LASSO-DCT at a noise level $\sqrt{\mathbb{E}\|\eta\|^2} = 0.1$. In each case, we choose values of the penalization parameter $\gamma$ to yield the best performance among the tested values. We use $\gamma = 0$ for Glow prior and $\gamma = 0.01$ for LASSO-WVT, and LASSO-DCT, respectively.

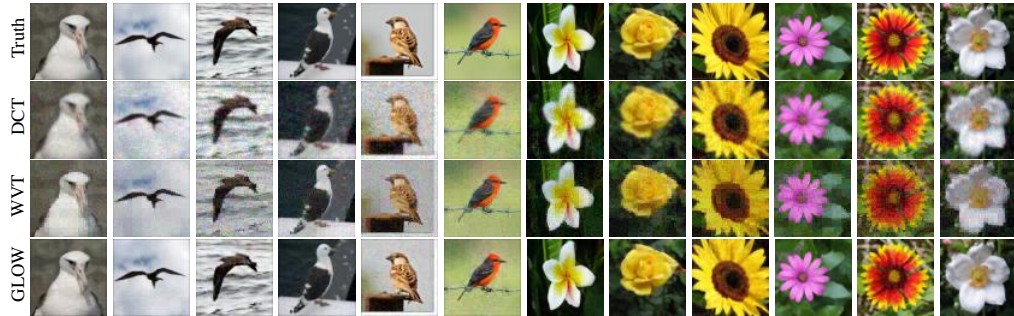

Figure 39: Compressed sensing — Visual comparisons on the test set images from Birds and Flowers dataset with a number $m = 5,000$ ($\approx 41\%$) of measurements under the Glow prior, LASSO-WVT, and LASSO-DCT at a noise level $\sqrt{\mathbb{E}\|\eta\|^2} = 0.1$. In each case, we choose values of the penalization parameter $\gamma$ to yield the best performance among the tested values. We use $\gamma = 0$ for Glow prior and $\gamma = 0.01$ for LASSO-WVT, and LASSO-DCT, respectively.

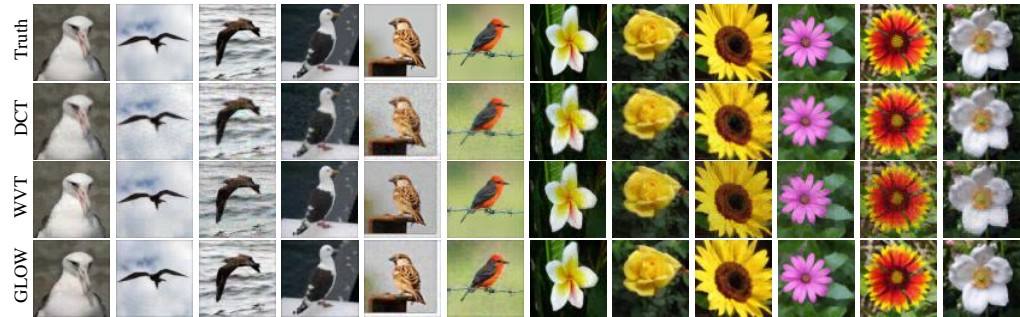

Figure 40: Visual comparisons of compressed sensing of the test set images from Birds and Flowers dataset with a number $m = 7,500$ ($\approx 61\%$) of measurements under the Glow prior, LASSO-WVT, and LASSO-DCT at a noise level $\sqrt{\mathbb{E}\|\eta\|^2} = 0.1$. In each case, we choose values of the penalization parameter $\gamma$ to yield the best performance among the tested values. We use $\gamma = 0$ for Glow prior and $\gamma = 0.01$ for LASSO-WVT, and LASSO-DCT, respectively.

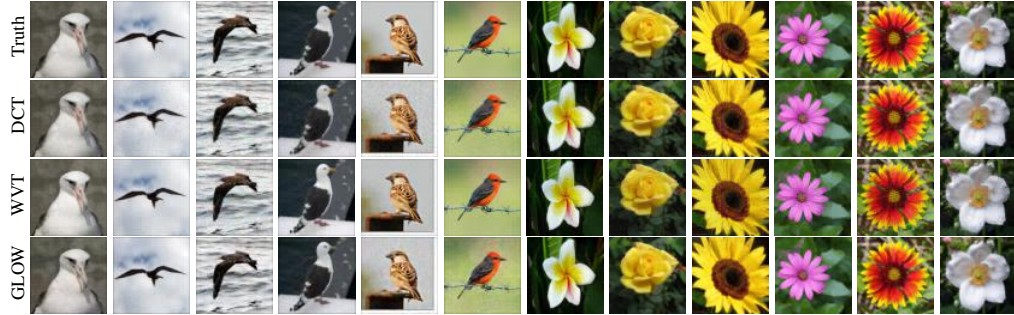

Figure 41: Visual comparisons of compressed sensing of the test set images from Birds and Flowers dataset with a number $m = 10,000$ ($\approx 81\%$) of measurements under the Glow prior, LASSO-WVT, and LASSO-DCT at a noise level $\sqrt{\mathbb{E}\|\eta\|^2} = 0.1$. In each case, we choose values of the penalization parameter $\gamma$ to yield the best performance among the tested values. We use $\gamma = 0$ for Glow prior and $\gamma = 0.01$ for LASSO-WVT, and LASSO-DCT, respectively.

## C.2 Compressed Sensing on Out of Distribution Images

Lack of representation error in invertible nets leads us to an important and interesting question: does the trained network fit related natural images that are underrepresented or even unrepresented in the training dataset? Specifically, can a Glow network trained on CelebA faces be a good prior on other faces; for example, those with dark-skin tone, faces with glasses or facial hair, or even animated faces? In general, our experiments show that Glow prior has an excellent performance on such out-of-distribution images that are semantically similar to celebrity faces but not representative of the CelebA dataset. In particular, we have been able to recover faces of darker skin tone, older people with beards, eastern women, men with hats, and animated characters such as Shrek, from compressed measurements under the Glow prior. Recoveries under the Glow prior convincingly beat the DCGAN prior, which shows a definite bias due to training. Not only that, the Glow prior also outperforms unbiased methods such as LASSO-DCT, and LASSO-WVT.

Can we expect the Glow prior to continue to be an effective proxy for arbitrarily out-of-distribution images? To answer this question, we tested arbitrary natural images such as car, house door, and butterfly wings that are semantically unrelated to CelebA images. In general, we found that Glow is an effective prior at compressed sensing of out-of-distribution natural images, which are assigned a high likelihood score (small normed latent representations). On these images, Glow also outperforms LASSO.

Recoveries of natural images that are assigned very low-likelihood scores by the Glow model generally run into instability issues. During training, invertible nets learn to assign high likelihood scores to the training images. All the network parameters such as scaling in the coupling layers of Glow network are learned to behave stably with such high likelihood representations. However, on very low-likelihood representations, unseen during the training process, the networks becomes unstable and outputs of network begin to diverge to very large values; this may be due to several reasons, such as normalization (scaling) layers not being tuned to the unseen representations. An LBFGS search for the solution of an inverse problem to recover a low-likelihood image leads the iterates into neighborhoods of low-likelihood representations that may lead the network to instability.

We find that Glow network has the tendency to assign higher likelihood scores to arbitrarily out-of-distribution natural images. This means that invertible networks have at least partially learned something more general about natural images from CelebA dataset — may be some high level features that face images share with other natural images such as smooth regions followed by discontinuities, etc. This allows Glow prior to extend its effectiveness as a prior to other natural images beyond just the training set.

Figure 42, 43 , 44, 45, and 46 compare the performance of LASSO-DCT, LASSO-WVT, DCGAN prior, and Glow prior on the compressed sensing of out-of-distribution images under varying number of measurements.

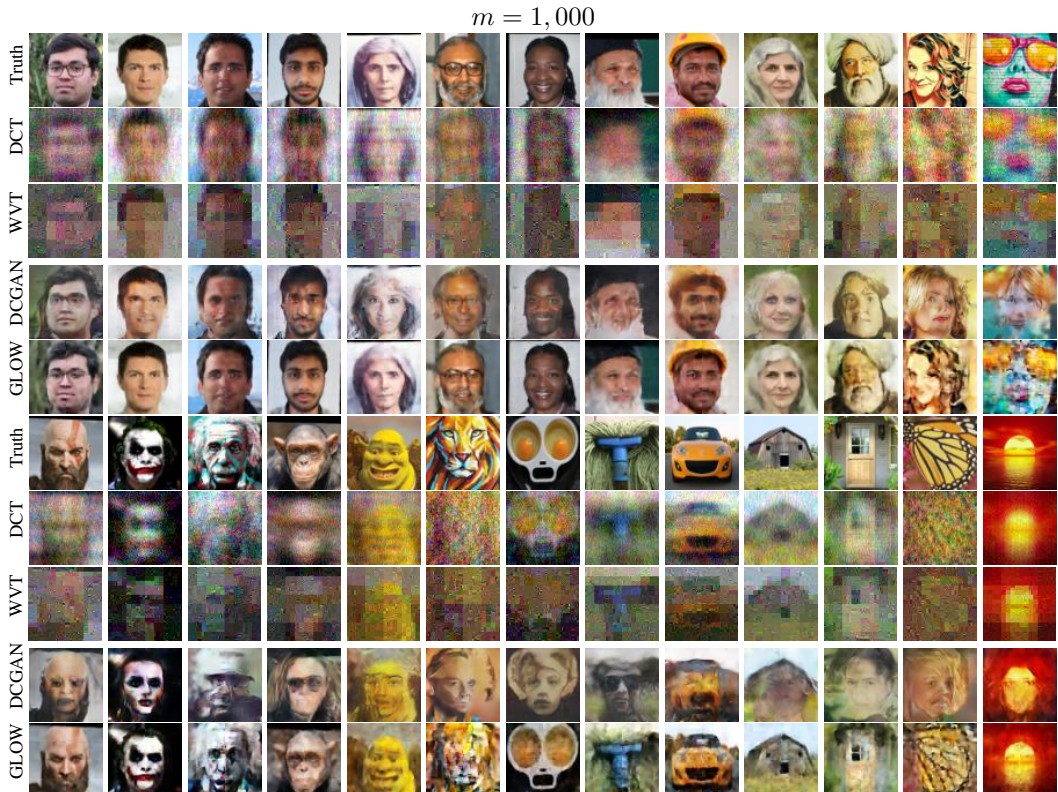

Figure 42: Compressed sensing ($m = 1000 \approx 8\%$ of $n$) visual comparisons on out-of-distribution images. We compare the recoveries under Glow (trained on CelebA) prior, DCGAN (trained on CelebA) prior, LASSO-WVT, and LASSO-DCT at a noise level $\sqrt{\mathbb{E}\|\eta\|^2} = 0.1$. In each case, we choose values of the penalization parameter $\gamma$ to yield the best performance. We use $\gamma = 0$ for both DCGAN, and Glow prior and and optimize $\gamma$ for each recovery using LASSO-WVT, and LASSO-DCT.

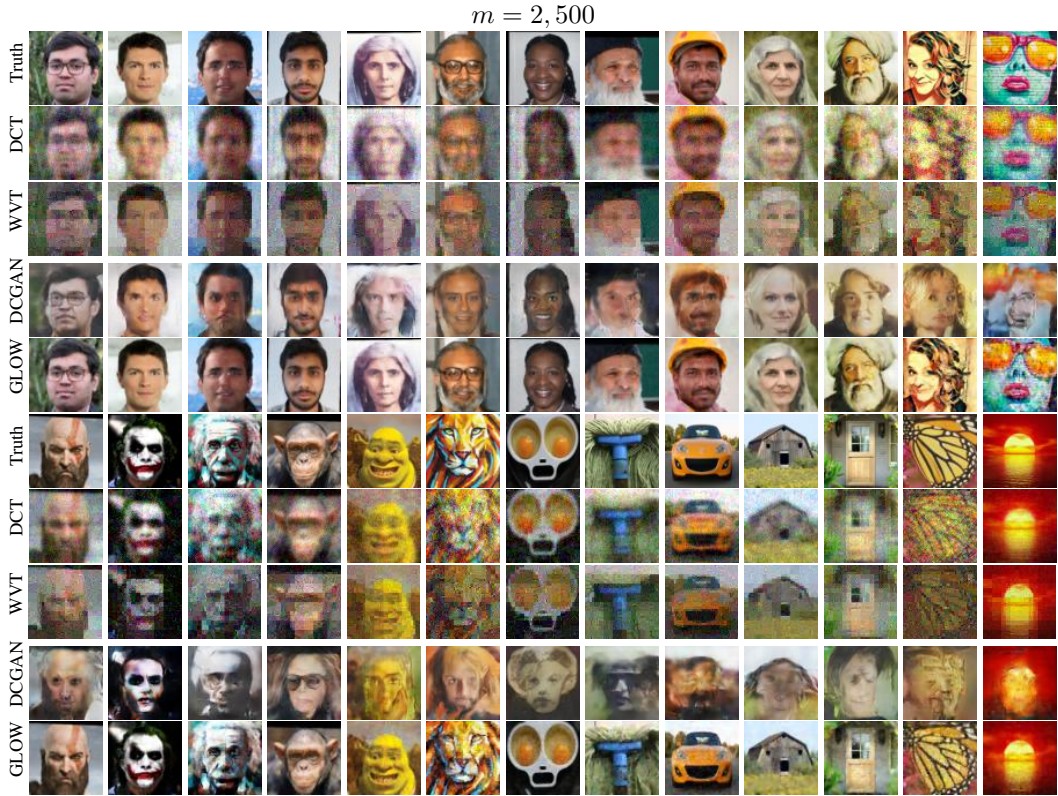

Figure 43: Compressed sensing ($m = 2500 \approx 20\%$ of $n$) visual comparisons on out-of-distribution images. We compare the recoveries under Glow (trained on CelebA) prior, DCGAN (trained on CelebA) prior, LASSO-WVT, and LASSO-DCT at a noise level $\sqrt{\mathbb{E}\|\eta\|^2} = 0.1$. In each case, we choose values of the penalization parameter $\gamma$ to yield the best performance. We use $\gamma = 0$ for both DCGAN, and Glow prior and and optimize $\gamma$ for each recovery using LASSO-WVT, and LASSO-DCT.

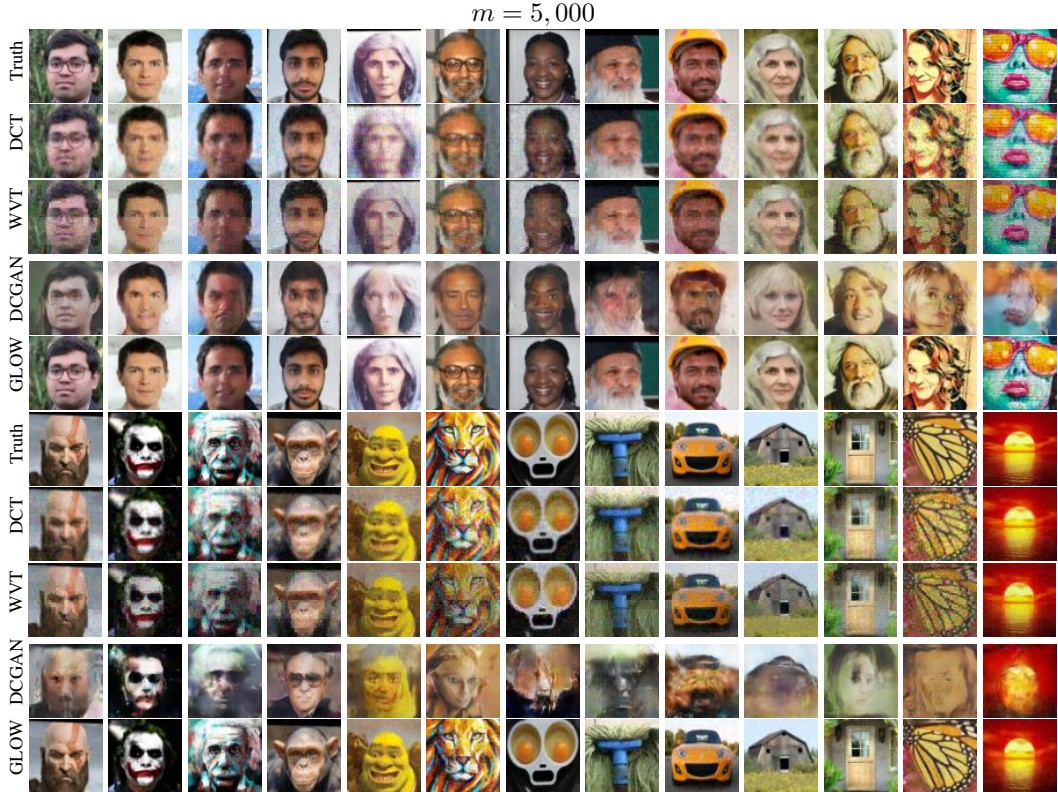

Figure 44: Compressed sensing ($m = 5000 \approx 41\%$ of $n$) visual comparisons on out-of-distribution images. We compare the recoveries under Glow prior (trained on CelebA), DCGAN prior (trained on CelebA), LASSO-WVT, and LASSO-DCT at a noise level $\sqrt{\mathbb{E}\|\eta\|^2} = 0.1$. In each case, we choose values of the penalization parameter $\gamma$ to yield the best performance. We use $\gamma = 0$ for both DCGAN, and Glow prior and and optimize $\gamma$ for each recovery using LASSO-WVT, and LASSO-DCT.

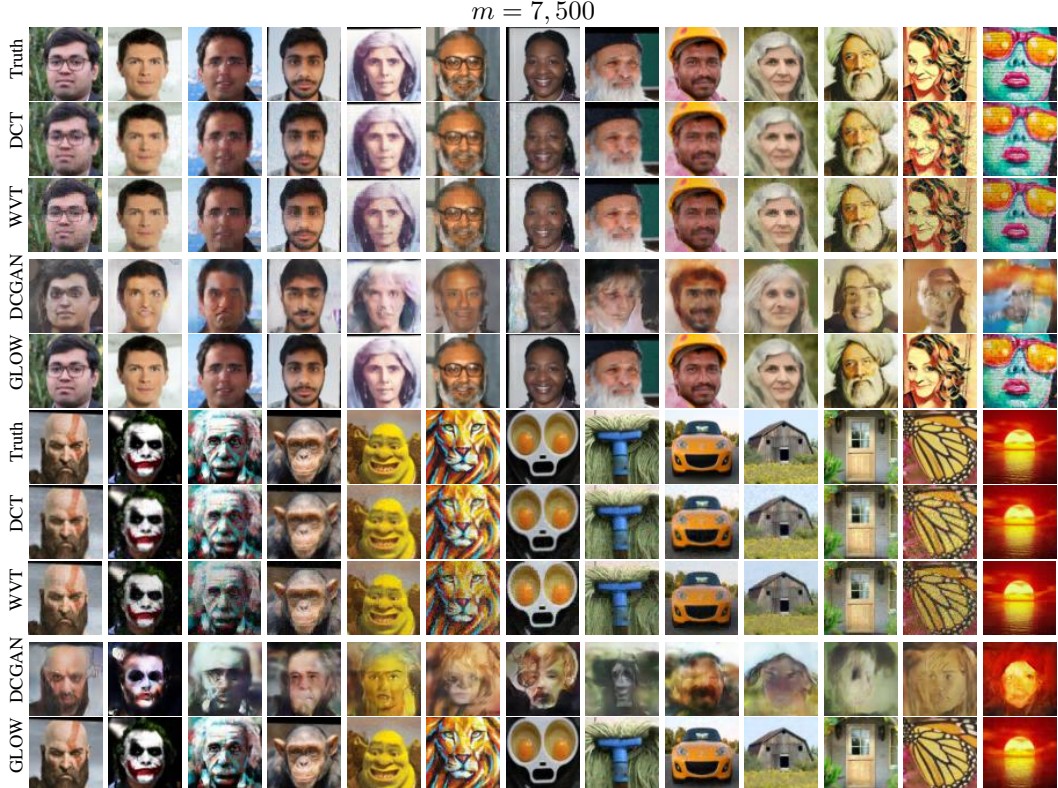

Figure 45: Compressed sensing ($m = 7500 \approx 61\%$ of $n$) visual comparisons on out-of-distribution images. We compare the recoveries under Glow prior (trained on CelebA), DCGAN prior (trained on CelebA), LASSO-WVT, and LASSO-DCT at a noise level $\sqrt{\mathbb{E}\|\eta\|^2} = 0.1$. In each case, we choose values of the penalization parameter $\gamma$ to yield the best performance. We use $\gamma = 0$ for both DCGAN, and Glow prior and and optimize $\gamma$ for each recovery using LASSO-WVT, and LASSO-DCT.

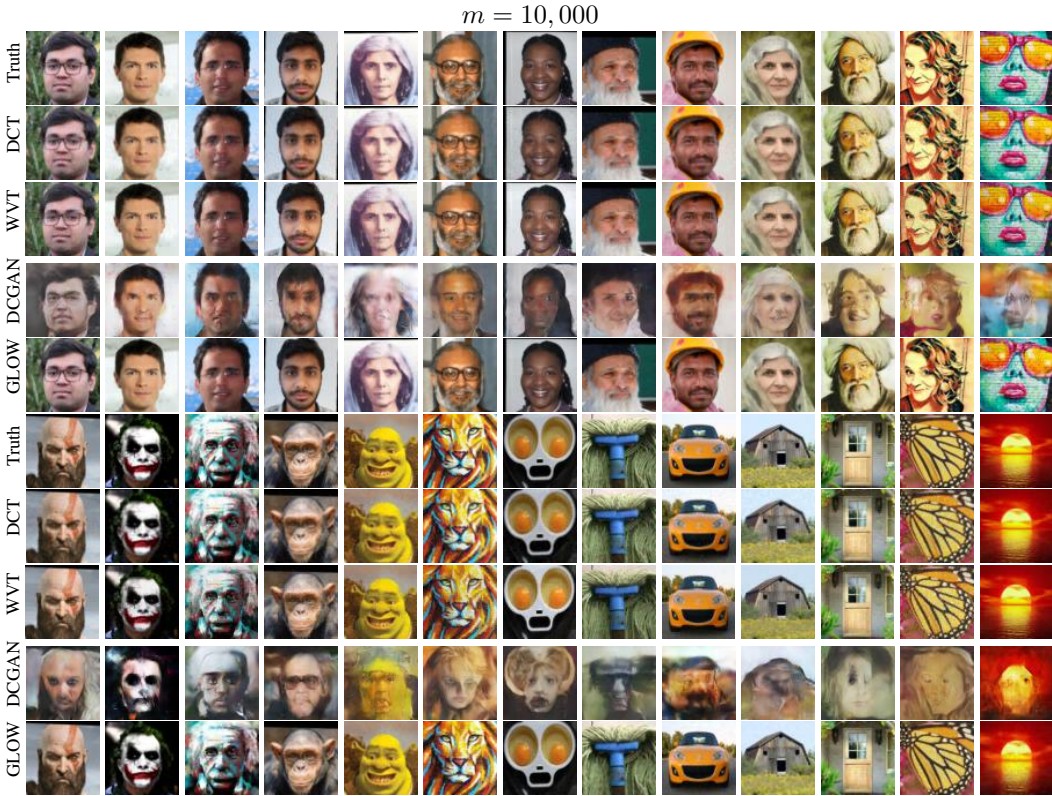

Figure 46: Compressed sensing ($m = 10,000, \approx 81\%$ of $n$) visual comparisons on out-of-distribution images. We compare the recoveries under Glow prior (trained on CelebA), DCGAN prior (trained on CelebA), LASSO-WVT, and LASSO-DCT at a noise level $\sqrt{\mathbb{E}\|\eta\|^2} = 0.1$. In each case, we choose values of the penalization parameter $\gamma$ to yield the best performance. We use $\gamma = 0$ for both DCGAN, and Glow prior and and optimize $\gamma$ for each recovery using LASSO-WVT, and LASSO-DCT.

# D    IMAGE INPAINITING

Our experiments with inpainting reveal a similar story as with compressed sensing. Compared to DCGAN, the recovered PSNRs using Glow prior are much higher under appropriate $\gamma$ as depicted in the right panel in Figure 47. If improperly initialized, then performance for $\gamma = 0$ could be poor. Even if improperly initialized, sufficiently large $\gamma$ leads to higher PSNRs.

As with compressive sensing, if the initialization is from a small latent variable, then the empirical risk formulation with $\gamma = 0$ exhibits high PSNRs. Algorithmic regularization is again occurring due to the small latent variable initialization.

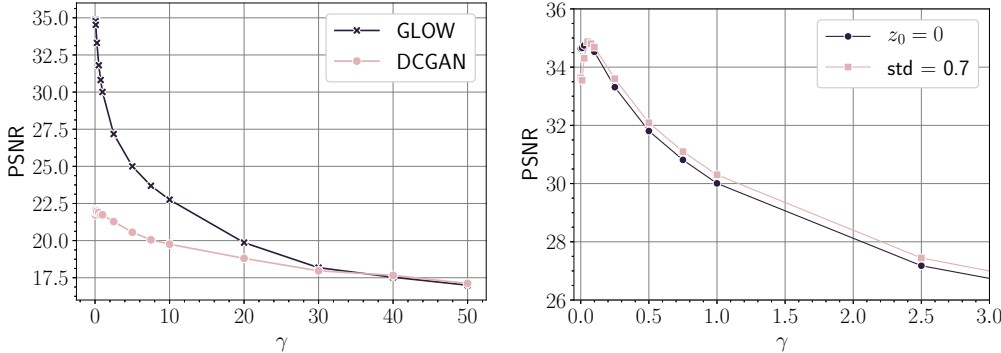

Figure 47: Inpainiting: PSNR (averaged over 12 test images of CelebA) vs. penalization parameter $\gamma$ under Glow prior and DCGAN prior (left panel) and using different initializations under Glow prior (right panel).

We present here qualitative results on image inpainting under the DCGAN prior, and the Glow prior on the CelebA test set. Compared to DCGAN, the reconstructions from Glow are of noticeably higher visual quality.

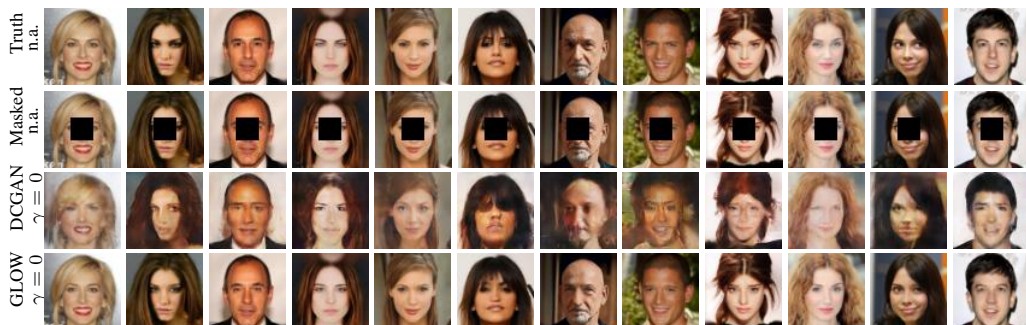

Figure 48: Image inpainting results on CelebA test set. Masked images are recovered under DCGAN prior and Glow prior. Recoveries under DCGAN prior are skewed and blurred whereas Glow prior leads to sharper and coherent inpainted images. For both Glow and DCGAN, we set $\gamma = 0$.

## D.1    IMAGE INPAINTING ON OUT OF DISTRIBUTION IMAGES

We now perform image inpainting under Glow prior, and DCGAN prior each trained on CelebA. Figure 49 shows the visuals of out-of-distribution inpainting. As before, DCGAN continues to suffer due to representation limits and data bias while Glow achieves reasonable reconstructions on out-of-distribution images semantically similar to CelebA faces. As one deviates to other natural images such as houses, doors, and butterfly wings, the inpainting performance deteriorates. At compressed sensing, Glow performed much better on such arbitrarily out-of-distribution images as good recoveries there only require the network only to assign a higher likelihood score to the true

image compared to the all the candidate static images given by the null space of the measurement operator.

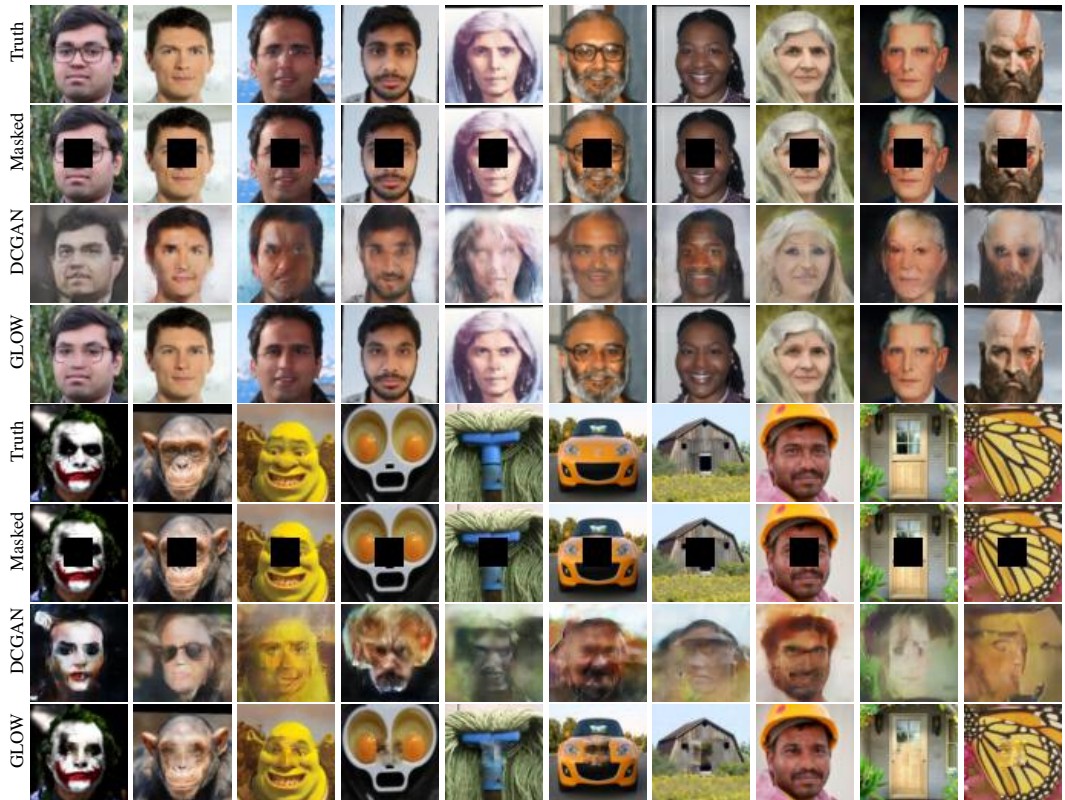

Figure 49: Image inpainiting results on out-of-distribution images. Masked images are recovered under DCGAN prior and Glow prior. Recoveries under DCGAN prior are skewed and blurred whereas Glow prior leads to sharper and coherent inpainted images. For both Glow and DCGAN, we set $\gamma = 0$.

## E    DISCUSSION

Figure 50 confirms the intuition brought up in the Discussion Section of the main paper that trained Glow network assigns lower likelihoods (larger latent representations) to noisy images. Histograms show that noisy images are generally occupy the less likelihood regimes or equivalently, the larger norm latent representations.

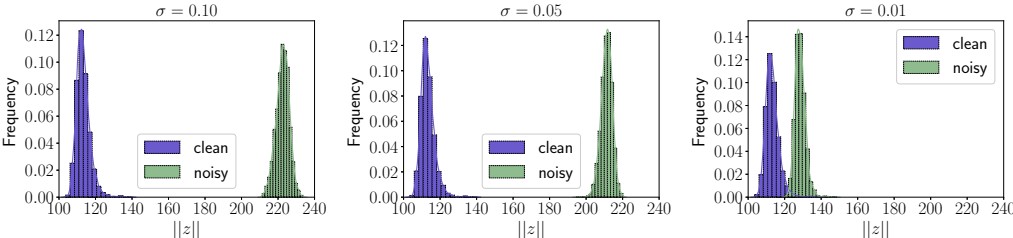

Figure 50: Histograms of the norm of the latent representation, $z$, over 3000 test images under additive Gaussian noise with $\sigma = 0.1$ (left), $\sigma = 0.05$ (middle), and $\sigma = 0.01$ (right).

Our experiments verify that natural images have smaller latent representations than unnatural images. Here we also show that adding noise to natural images increases the norm of their latent representations, and that higher noise levels result in larger increases. Additionally we provide evidence that random perturbations in image space induce larger changes in $z$ than comparable natural perturbations in image space. Figure 51 shows a plot of the norm of the change in image space, averaged over 100 test images, as a function of the size of a perturbation in latent space. Natural directions are given by the interpolation between the latent representation of two test images. For the denoising problem, this difference in sensitivity indicates that the optimization algorithm might obtain a larger decrease in $\|z\|$ by an image modification that reduces unnatural image components than by a correspondingly large modification in a natural direction.

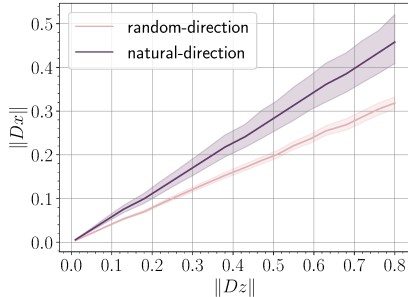

Figure 51: The magnitude of the change in image space as a function of the size of a perturbation in latent space. Solid lines are the mean behavior and shaded region depicts $95\%$ confidence interval.

## F    LOSS LANDSCAPE: DCGAN VS. GLOW

In Figure 52, we plot $\|y - AG(z^* + \alpha\delta_v + \beta\delta_w)\|^2$ versus $(\alpha, \beta)$ where $\delta_v$ and $\delta_w$ are scaled to have the same norm as $z^*$, the latent representation of a fixed test image. For DCGAN, we plot the loss landscape versus two pairs of random directions. For Glow, we plot the loss landscape versus a pair of random directions and a pair of directions that linearly interpolate in latent space between $z^*$ and another test image.

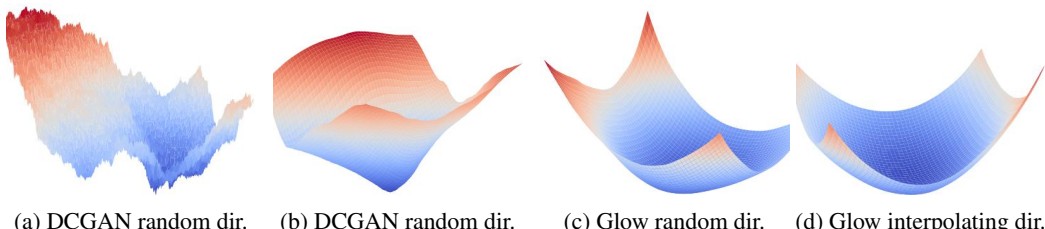

(a) DCGAN random dir.    (b) DCGAN random dir.    (c) Glow random dir.    (d) Glow interpolating dir.

Figure 52: Loss landscapes for $\|AG(z) - y\|_2^2 + \gamma\|z\|_2$ with $\gamma = 0$ around the latent representation of a fixed image and with respect to either random latent directions or latent directions that interpolate between images.

## G    IMAGE AND LATENT SPACE FORMULATIONS

As mentioned in the main paper, a natural formulation of the inverse problem is

$$\min_{x \in \mathbb{R}^n} \|Ax - y\|^2 - \gamma \log p_G(x), \tag{3}$$

where $p_G(x)$ is the target density. We instead formulate the inverse problem as

$$\min_{z \in \mathbb{R}^n} \|AG(z) - y\|_2^2 + \gamma\|z\|_2; \tag{4}$$

a measurement misfit combined with a Gaussian prior on the latent space.

We will denote the target distribution by $p_G(x)$ and the latent Gaussian distribution by $p(z)$. To illustrate the differences between equation 3 and equation 4, we train a Real-NVP model Dinh et al. (2016) on a synthetic two-dimensional dataset, visualize both the $\log p_G(x)$ and $\log p(z)$ in latent and image space, and solve a simple compressive sensing recovery problem. Our two dimensional data points are generated by sampling the first coordinate $x_1$ from a bimodel Gaussian distribution and the second coordinate $x_2$ from a uniform distribution as shown in Figure 53.

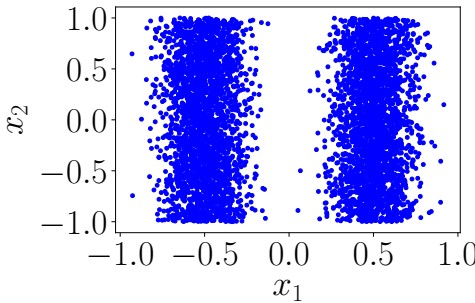

Figure 53: A point cloud of the synthetically generated data $x \in \mathbb{R}^2$

For comparison, we plot the $x$-likelihood versus $x$ (left), latent $z$-likelihood versus $x$ (middle), and $x$-likelihood versus $z$ (right) in Figure 54. These plots illustrate that generally high-likelihood $x$ points are also given higher latent $z$-likelihood, however, some low $x$-likelihood might be assigned a higher Gaussian $z$-likelihood; these are, for example, the points living on the darker contour spearing through the Gaussian bowl in the right plot. Figure 55 shows some of the points in the $x$-likelihood (left) that map to this contour in the $z$-space (right).

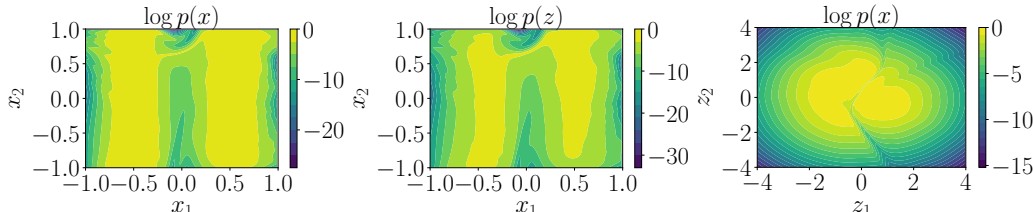

Figure 54: $x$-likelihood versus $x$ (left), $z$-likelihood versus $x$ (middle), and $x$-likelihood versus $z$ (right).

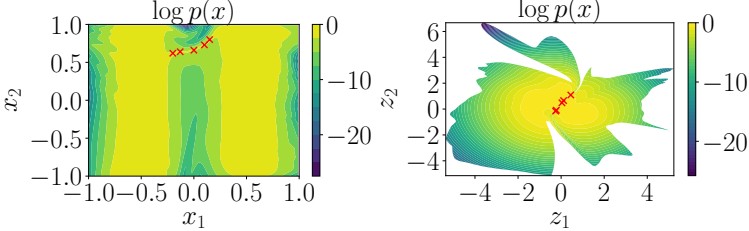

Figure 55: Some points (red-crosses) in $x$-space mapped to $z$-space. The (unwanted, as it corresponds to low-likelihood points) bridge connecting the models of the learned bimodal distribution is mapped to the contour in the $z$-space.

### G.1 COMPRESSIVE SENSING IN 2D

To compare latent-space formulation equation 4 and data-space formulation equation 3, we construct a simple compressive sensing recovery problem for this two-dimensional data and illustrate the

difference under both good and bad initializations. Specifically, we want to recover a vector $x = [x_1 \ x_2]^\mathsf{T}$ from a single linear measurement $y = \langle a, x \rangle = x_2$, where $a = [0 \ 1]^\mathsf{T}$. Figure 56 shows the gradient descent path, and final solution, while solving equation 4 (left column), and equation 3 (right column) from a good and a bad initialization. $x$-likelihood formulation seems more robust to a bad initialization in this case compared to $z$-likelihood as $z$-likelihood might not be a good proxy for $x$-likelihood for some points. This bad case is carefully crafted to illustrate the difference between the two formulations, however, in practice, it seems unlikely that a low $x$-likelihood points that somehow achieves higher $z$-likelihood will also obey the measurement constraints.

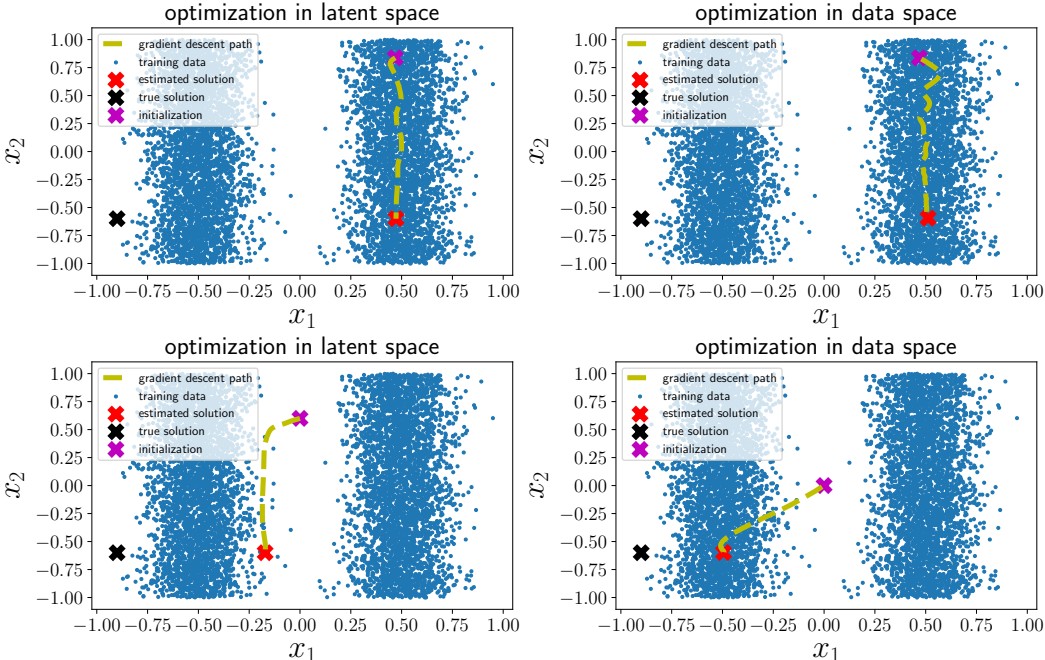

Figure 56: We show gradient descent path from the initialization to the final estimate along with true solution. In the first row, we initialized from $z = 0$ (good initialization) and in the second row we used low likelihood data points as intializations (bad initialization).

## G.2 COMPRESSIVE SENSING FOR CELEBA

In case of CelebA images, we found that optimizing over direct likelihood of images proved very hard to tune. To better understand why equation 4 is easier compared to equation 3, we draw the landscape of the loss surfaces of equation 4 versus $z$ and equation 3 versus $x$ under different $\gamma$ in two random directions around an the ground truth in $z$, or $x$, as appropriate; see Figure 57. In the $x$-formulation the loss surfaces (first row) have a sharp dip at the ground truths, which comes from $-\log p_G(x)$ term. We believe that sharp dip in the loss surface makes it difficult to tune the $\gamma$ parameter, the learning rate, and makes the optimization using equation 3 numerically more challenging as observed in our experiments. On the other hand, the loss surfaces for equation 4 (second row) appear smoother.

We now show a quantitative comparison of the $x$-likelihood formulation in equation 3, and $z$-likelihood formulation in equation 4 on compressive sensing for CelebA test images versus $m$ for fixed values of $\gamma$; see Figure 58. We initialize with $z_0 = 0$, and $x_0 = G(z_0)$, as appropriate. We simply choose $\gamma = 0$ in equation 4. However, we need to choose $\gamma$ more carefully in equation 3, and different values of $\gamma$ are appropriate across different undersampling ratios. Even if one ignores the difficulty of choosing the hyperparameter $\gamma$, the formulation in equation 4 generally performs much better than equation 3 as evident from the plots.

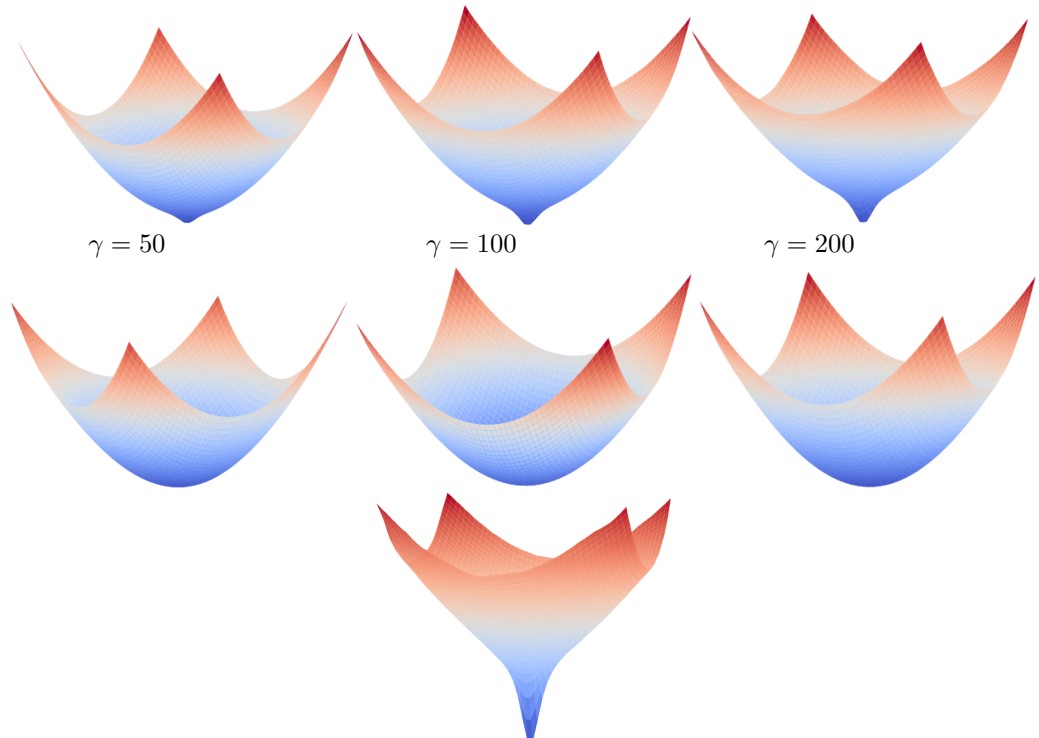

Figure 57: Landscapes of the loss surfaces in the $x$-space (first row), the loss surfaces of in the $z$-space (second row) for various values of $\gamma$, and loss surface of $x$-likelihood $-\log p(x)$ (third row).

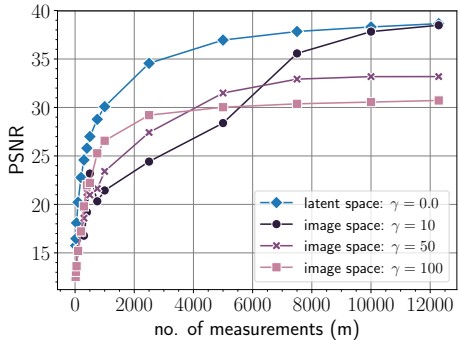

Figure 58: We report PSNR against number of measurements $m$ when optimizing in the latent space equation 4 with $\gamma = 0$ and the image space equation 3 with $\gamma$ set to 10, 50 and 100.

To show the effect of noise on recovery in compressive sensing under different values of $\gamma$ and noise levels, we plot PSNR of the iterates when solving equation 4 against iterations in Figure 59. This plot shows, perhaps surprisingly, that even under noisy compressed measurements it is a good idea to solve the inverse compressed sensing problem equation 4 with $\gamma = 0$.

### G.3 DENOISING FOR CELEBA

For completeness, we also compare denoising using our latent space formulation equation 4, our image space forumation equation 3 under different noise levels $\sigma = 0.05$ and $\sigma = 0.10$; see Figure 60 and Figure 61 respectively. For both noise levels, we observe equal performance (indicated by the highest PSNR) when optimizing in the latent or image space. We do not report results over $\sigma = 0.20$ as it was hard to tune hyper paremeters for higher noise levels in equation 3.

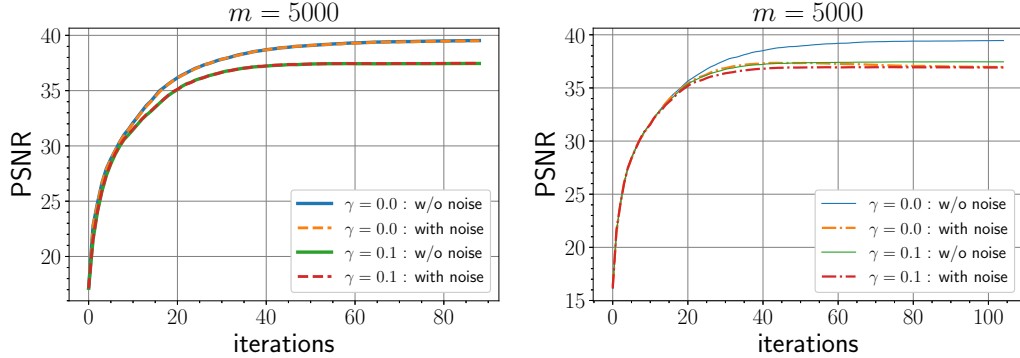

Figure 59: We plot PSNR against gradient iterations for compressive sensing at $m = 5000$ on a single image under the presence and absence of noise with different values of $\gamma$ with noise level $\sqrt{\mathbb{E}\|\eta\|^2} = 0.1$ (left) and $\sqrt{\mathbb{E}\|\eta\|^2} = 1$ (right).

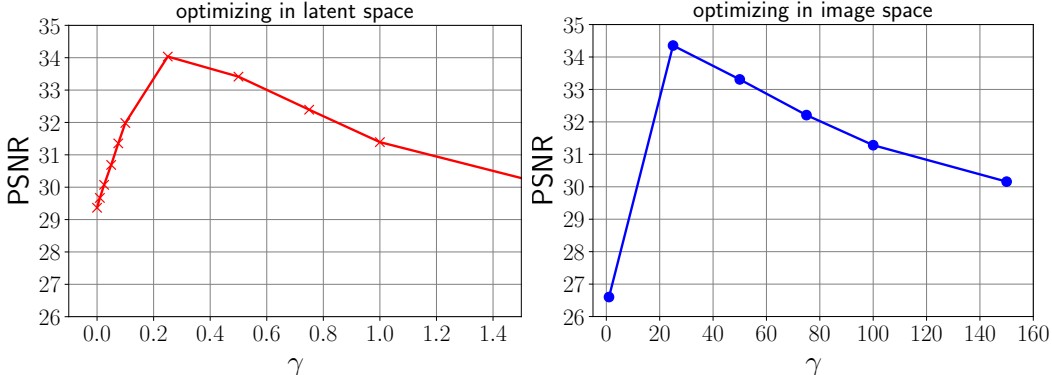

Figure 60: Denoising comparision at $\sigma = 0.05$ when optimizing over latent space (left) versus image space (right).

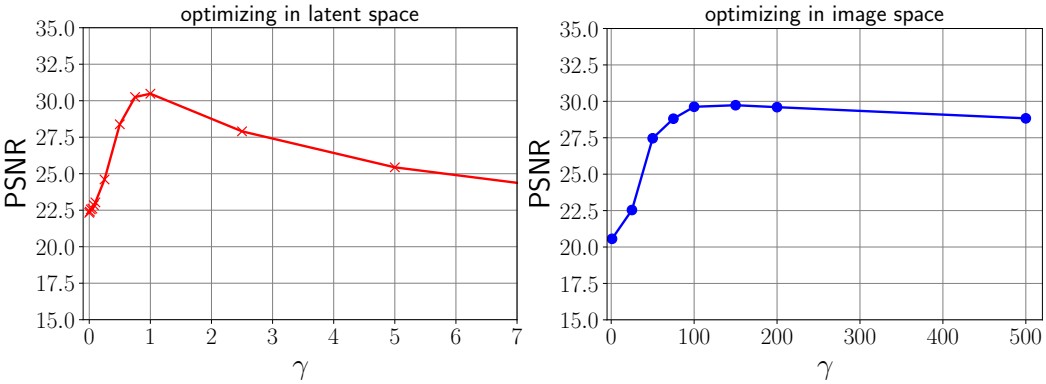

Figure 61: Denoising comparision at $\sigma = 0.10$ when optimizing over latent space (left) versus image space (right).

