# OpenReview forum: "Invertible generative models for  inverse problems: mitigating representation error and dataset bias"
_ICLR.cc/2020/Conference — Reject_

### Official Review · AnonReviewer3 · 2019-10-24
**Official Blind Review #3**

**Rating:** 6

**Review:**

Update: I have read the other reviews and the author response and have not changed my evaluation.

Recent work has shown that GANs can be effective for use as priors in inverse problems for images such as compressed sensing, denoising, and inpainting. A drawback is that GANs may have the problem of inexact reconstruction, and strongly reflect the biases in the training set yielding poor performance on out-of-distribution data. This paper shows that the exact inverses available to normalizing flow models and their broad assignment of likelihoods allows for better reconstructions especially on out-of-domain data.

As far as I know, this is the first work to use normalizing flows for inpainting and compressed sensing. The approach and application is very natural, although it’s a bit surprising that using the likelihood as a prior term directly did not work very well. The results of this work show that invertible generative models have utility for inverse image problems even when the quality of raw samples is substantially below GANs. In my opinion the main advantage in this method is not on having low reconstruction error on observed pixels, which becomes less of a problem for more powerful GAN models, but rather the good performance on out of domain data which is somewhat surprising. The authors are reasonably thorough, testing their model on a variety of problem settings and perform ablation studies on hyperparameters.

As additional baselines for compressed sensing and denoising, it would be good to compare to the Deep Image Prior since there is effectively no out-of-distribution input for this untrained model and it performs well with moderate image corruption. Additional discussion about the two could be useful, as for the Deep Image Prior a similar patter is observed where denoising requires explicit regularization (early stopping or gradient noise for DIP) but image completion and compressed sensing do not. Also, there have been many improvements to DCGAN over the years that might ameliorate the problems that were observed in reconstruction, but I don’t fault the authors much for this as it can be difficult training models like StyleGAN even at 64x64 sizes.

It might also be interesting to know whether the good performance on out-of-distribution inputs is due to the exact invertibility or the log-likelihood objective, although I would guess that it is the latter. On way to test this would be training the GLOW model with an adversarial objective instead of NLL as done in [1].

Minor Comments:

Figure 4 would probably be better with a logarithmic scaling for # of measurements

I did not understand the comment about sublevel sets of the data misfit term being inverse images of cylinders, maybe this could use some elaboration.

[1] https://arxiv.org/abs/1705.08868


**Experience Assessment:**

I have published one or two papers in this area.

**Review Assessment: Checking Correctness Of Derivations And Theory:**

N/A

**Review Assessment: Checking Correctness Of Experiments:**

I carefully checked the experiments.

**Review Assessment: Thoroughness In Paper Reading:**

I read the paper at least twice and used my best judgement in assessing the paper.

---

> ### Author Response · Authors · 2019-11-14
> **Comparison to Unlearned Methods and Other Clarifications**
>
> Thank you for your thorough reading of the paper.  We agree that a comparison to Deep Image Prior is quite interesting.  This has actually been done in another paper submitted to this conference [https://openreview.net/pdf?id=rkegcC4YvS].  Technically, they compare the DCGAN to the Deep Decoder, which is an underparameterized Deep Image Prior with simpler architecture and comparable performance.  Figure 1 (lower panels) in that paper demonstrate that the Deep Decoder underperforms the DCGAN when there are few enough measurements (m<500 in the same problem size as our experiments).  Also that figure shows the Deep Decoder gives consistently lower PSNRs than we report for INNs across the entire range of undersampling ratios.  In the case of significantly under sampled measurements, the Deep Decoder is over 5 dB worse in PSNR.  We will add a remark to this effect in the camera ready, if accepted.
>
> It is a great suggestion to determine if out-of-distribution performance is due to invertibility or log-likelihood optimization.  A good argument can be made on both sides.  We are attempting to do this in time for the camera-ready.  Getting adversarial training to converge is challenging, but we will try and will add a remark to the camera ready.
>
>
> We have clarified the comment about the sublevel sets.  The remark was intended to say that because of invertibility of the model G, there are no local minima (aside from global minima) of the data misfit term in z-space (||A G(z) - y||^2).  This ensures that the latent optimization we propose has a favorable landscape for convergence.

---

### Official Review · AnonReviewer1 · 2019-10-24
**Official Blind Review #1**

**Rating:** 1

**Review:**

This paper proposes to employ the likelihood of the latent representation of images as the optimization target in the Glow (Kingma and Dhariwal, 2018) framework. The authors argue that to optimize the ''proxy for image likelihood'' has two advantages: First, the landscapes of the surface are more smooth; Second, a latent sample point in the regions that have a low likelihood is able to generate desired outcomes. In the experimental analysis, the authors compare their proposed method with several baselines and show prior performance.

This paper has three major flaws and should be clearly rejected.
First, the novelty of this paper is trivial, in my opinion, the Eq. 2 is the only contribution of this paper.
Second, the experimental results are not convincing, almost all the methods proposed after 2015 have better performance compared to these baseline methods.
Third, there are a lot of claims in this paper have been made without clarification, I have huge troubles in understanding certain sentences.


**Experience Assessment:**

I have read many papers in this area.

**Review Assessment: Checking Correctness Of Derivations And Theory:**

N/A

**Review Assessment: Checking Correctness Of Experiments:**

I carefully checked the experiments.

**Review Assessment: Thoroughness In Paper Reading:**

I read the paper thoroughly.

---

> ### Author Response · Authors · 2019-11-14
> **Strong empirical results under a principled new framework for inversion**
>
> Thank you for the thoughtful comments.
>
> Novelty of the paper:  The primary novelty of this paper is the proof of concept that invertible neural networks (INNs), out of the box, are surprisingly effective image priors for inverse problems, especially on out-of-distribution images.  This behavior was not known before this paper and can not be found in the previous literature either on invertible neural networks or on the literature in signal recovery.  There was one paper that uses INNs to directly learn a specific forward map, returning the inverse for free, but this method would need to be retrained for every variation of every inverse problem.   As a result of our paper, a practitioner who is building an image prior for a given distribution class aught to carefully consider the option of training an invertible net on their desired signal class.  (Naturally, they should also consider other methods too in order to see what works best for their problem).  Without this work, it is likely one might not think to give INNs a try because they are a substantially different architecture than everything else in the literature.  Other novelties of the paper are: in denoising, we introduce a formulation that directly optimizes image likelihood (demonstrating the strength of INNs in density estimation), and in compressed sensing we introduce formulation (2), which surprisingly works best with no direct likelihood penalization (gamma=0).
>
> Comparison to other methods: The purpose of this paper is to point out the promise of an entirely different framework for generative image priors for inverse problems.  We specifically did not put bells and whistles on the Invertible Neural Networks we trained because we wanted to show how much better the out-of-the-box performance of INNs was compared to GANs. Much followup work has happened with GANs in an attempt to lower their representation error.  These include Image Adaptive GANs and Latent Convolutional Models, which both use ideas from untrained neural networks (such as optimizing the weights of a neural network at inversion time).  Similar ideas could also be used for our Invertible Neural Network models, which will similarly make their level of performance even greater.
>
> Clarity of paper:  If accepted, at the camera ready, we will clarify any sentences that the reviewer finds unclear.  We will also seek additional eyes before the camera ready in order to identify which sentences need additional clarity.

---

### Official Review · AnonReviewer2 · 2019-10-30
**Official Blind Review #2**

**Rating:** 3

**Review:**

Authors extend the invertible generative model of Kingma et. al. to image inverse problems. Specifically, they use Generator trained within the Glow framework as an image prior for de-noising, inpainting and compressed sensing tasks.
During training, a heuristic adjustment to the objective is made allowing optimization of latent variable norm instead of image log likelihood. This seemed critical for convergence of image inverse tasks. The use of Glow prior was shown to be beneficial for all inverse tasks. Experiments were limited to face images from celebA database. While the proposal demonstrates improved empirical performance, it seems to be the only contribution of this paper. Taking an existing model and applying it to a problem where similar extensions have been tried (GAN etc) does not seem quite worthy of a full paper.

**Experience Assessment:**

I have published in this field for several years.

**Review Assessment: Checking Correctness Of Derivations And Theory:**

I assessed the sensibility of the derivations and theory.

**Review Assessment: Checking Correctness Of Experiments:**

I did not assess the experiments.

**Review Assessment: Thoroughness In Paper Reading:**

I read the paper thoroughly.

---

> ### Author Response · Authors · 2019-11-14
> **Strong empirical results under a principled new framework for inversion**
>
> Thank you for the thoughtful comments.
>
> Worthiness of being a full paper:  We argue that this work is worthy of being a full paper because it shows impressive empirical results for a principled signal recovery paradigm that address the central challenge facing generative models as image prior.  That central challenge is representation error, and our principled solution is invertible neural networks which can be trained by directly optimizing likelihood of test images.  The out-of-distribution performance we report is particularly important:  if one were to train a GAN for MRI images, it will be impossible to ensure all possible pathologies are in the training set, and thus images must generalize beyond their training data, which invertible nets do.  Given the substantial costs of using INNs, this paper provides a significant contribution to the field by demonstrating feasibility of an approach that might initially appear unlikely to work.

---

### Official Review · AnonReviewer4 · 2019-10-31
**Official Blind Review #4**

**Rating:** 6

**Review:**

This paper investigates the performance of invertible generative models for solving inverse problems. They argue that their most significant benefit over GAN priors is the lack of representation error that (1) enables invertible models to perform well on out-of-distribution data and (2) results in a model that does not saturate with increased number of measurements (as observed with GANs). They use a pre-trained Glow invertible network for the generator and solve a proxy for the maximum likelihood formulation of the problem, where the likelihood of an image is replaced by the likelihood of its latent representation. They demonstrate results on problems such as denoising, inpainting and compressed sensing. In all these applications, the invertible network consistently outperforms DCGAN across all noise levels/number of measurements. Furthermore, they demonstrate visually reasonable results on natural images significantly different from those in the training dataset.

The idea of using invertible networks for estimating a specific forward process is not new, as the authors also pointed out. The contribution of this paper is that they use a pre-trained invertible model as a prior in various tasks not known in training time and support their technique with experimental results and therefore I would recommend accepting this paper.

Since one of the main arguments in the paper is how the lack of representation error benefits the Glow prior compared to DCGAN prior, it would be interesting to see the representation error quantitatively for the DCGAN results and how it contributes to the total error.  Moreover, demonstrating the comparison results in other metrics than PSNR (MSE, SSIM) would be interesting and more comprehensive.

**Experience Assessment:**

I have read many papers in this area.

**Review Assessment: Checking Correctness Of Derivations And Theory:**

N/A

**Review Assessment: Checking Correctness Of Experiments:**

I assessed the sensibility of the experiments.

**Review Assessment: Thoroughness In Paper Reading:**

I read the paper at least twice and used my best judgement in assessing the paper.

---

> ### Author Response · Authors · 2019-11-14
> **Thoughts on Representation Error and Other Metrics**
>
> Thank you for the careful reading of the paper. We agree that it is indeed quite interesting to see where are the sources of error in the DCGAN prior.  We did not include this in the paper because it is already included in the Bora et al. paper.  In Section 6.3 of their paper, they show that the dominant source of error is representation error (as opposed to measurement error or optimization error).  We have added a remark in Section 3.2 to this effect in the paper so that other readers can be aware of this observation.
>
> If the paper is accepted, in the camera ready supplemental material, we intend to include plots of recovery performance as measured by SSIM for the denoising and compressed sensing problems.  We already have presented the results in the MSE metric in the supplemental materials.

---

### Decision · Program_Chairs · 2019-12-19

**Decision:**

Reject

**Comment:**

This paper studies the empirical performance of invertible generative models for compressive sensing, denoising and in painting. One issue in using generative models in this area has been that they hit an error floor in reconstruction due to model collapse etc i.e. one can not achieve zero error in reconstruction. The reviewers raised some concerns about novelty of the approach and thoroughness of the empirical studies. The authors response suggests that they are not claiming novelty w.r.t. to the approach but rather their use in compressive techniques. My own understanding is that this error floor is a major problem and removing its effect is a good contribution even without any novelty in the techniques. However,  I do agree that a more thorough empirical study would be more convincing. While I can not recommend acceptance given the scores I do think this paper has potential and recommend the authors to resubmit to a future venue after a through revision.